# Randomized Primal-Dual Coordinate Method for Large-scale Linearly Constrained Nonsmooth Nonconvex Optimization

## Abstract

The large-scale linearly constrained nonsmooth nonconvex optimization finds wide applications in machine learning, including non-PSD Kernel SVM, linearly constrained Lasso with nonsmooth nonconvex penalty, etc. To tackle this class of optimization problems, we propose an efficient algorithm called Nonconvex Randomized Primal-Dual Coordinate (N-RPDC) method. At each iteration, this method only randomly selects a block of primal variables to update rather than updating all the variables, which is suitable for large-scale problems. We provide two types of convergence results for N-RPDC. We first show that any cluster point of the sequence of iterates generated by N-RPDC is almost surely (i.e., with probability 1) a stationary point. In addition, we also provide an almost sure asymptotic convergence rate of $O(1/\sqrt{k})$. Next, we establish the expected $O(\varepsilon^{-2})$ iteration complexity of N-RPDC in order to drive a natural stationarity measure below $\varepsilon$ in expectation. The fundamental aspect to establishing the aforementioned convergence results is a *surrogate stationarity measure* we discovered for analyzing N-RPDC. Finally, we conduct a set of experiments to show the efficacy of N-RPDC.

## 1 Introduction

Many large scale problems arising in machine learning amounts to solving the following linearly constrained nonsmooth nonconvex optimization problem:

$$\begin{aligned} \min_{x \in \mathbb{X}} \quad & F(x) = f(x) + g(x) \\ \text{s.t} \quad & Ax - b = 0, \end{aligned} \tag{P}$$

where $A \in \mathbb{R}^{n \times d}$ and $b \in \mathbb{R}^n$. Throughout this paper, we impose the following *assumptions* on problem (P): the function $f : \mathbb{R}^d \to \mathbb{R}$ is possibly nonconvex and continuously differentiable with its gradient $\nabla f$ being $L_f$-Lipschitz continuous, $\mathbb{X}$ is a convex and compact set of $\mathbb{R}^d$, i.e., there exits a positive number $M$ such that $M = \max_{x,x' \in \mathbb{X}} \|x - x'\|$, and $g : \mathbb{R}^d \to \mathbb{R}$ is nonsmooth and nonconvex. More precisely, $g$ is assumed to be lower semicontinuous (l.s.c) $\rho_g$-weakly convex and has bounded subgradients over $\mathbb{X}$, i.e., there exists $L_g > 0$ such that $\|s\| \le L_g, \forall s \in \partial g(x), x \in \mathbb{X}$, where $\partial g$ is the subdifferential of $g$ (see (4) for definition). Let $F^*$ be the optimal value of (P).

Recall that a function $g$ is said to be $\rho_g$-weakly convex if $g(\cdot) + \frac{\rho_g}{2} \|\cdot\|^2$ is convex for some constant $\rho_g \ge 0$ (Vial, 1983). It is worth mentioning that a wide class of nonsmooth nonconvex functions belong to the weakly convex class; see, e.g., (Vial, 1983; Davis & Drusvyatskiy, 2019) for more discussions on weak convexity.

We further assume $g(x) = \sum_{i=1}^N g_i(x_i)$ is block separable with respect to the space decomposition

$$\mathbb{X} = \mathbb{X}_1 \times \mathbb{X}_2 \times \cdots \times \mathbb{X}_N, \quad x_i \in \mathbb{X}_i \subset \mathbb{R}^{d_i}, \quad \text{and} \sum_{i=1}^N d_i = d. \tag{1}$$

Let $A = (A_1, A_2, \ldots, A_N) \in \mathbb{R}^{n \times d}$ be the corresponding partition of $A$, where $A_i \in \mathbb{R}^{n \times d_i}$. When $N = d$, each $\mathbb{X}_i \subset \mathbb{R}$ corresponds to a box constraint on the coordinate $x_i$ for $i = 1, \ldots, d$.

**Motivations.** Our motivation for studying large scale problem (P) stems from the fact that this problem class finds wide applications in machine learning. To be more specific, we will present several practical examples below that give rise to (P).

Application 1: Non-PSD kernel support vector machine. Support vector machine (SVM) is a widely utilized technique for supervised learning (Boser et al., 1992; Cortes & Vapnik, 1995). In order to improve the interpretability and enhance the robustness, some non-PSD kernels are used in the kernalized SVM method, such as the sigmoid kernel (Lin & Lin, 2003), the jittering kernel (DeCoste & Schölkopf, 2002), the tangent distance kernel (Haasdonk & Keysers, 2002, August), to name a few. Formally, the kernalized SVM problem can be written as

$$\min_{x \in [0,c]^d} \quad \tfrac{1}{2} x^\top Q x - \mathbf{1}_d^\top x$$
$$\text{s.t.} \quad y^\top x = 0, \tag{2}$$

where $Q$ is a $d \times d$ non-PSD matrix, $c \in \mathbb{R}$ is the upper bound of all variables, $y \in \{-1, 1\}^d$ is the vector of labels, and $\mathbf{1}_d$ is an $d$-dimensional vector of all 1s.

Application 2: Linearly constrained Lasso with nonsmooth nonconvex penalty. The Lasso is one of the most popular methods for variable selection. The standard Lasso uses $\ell_1$-norm to select important variables. Some works propose to utilize nonsmooth nonconvex regularizers to further improve the performance of the Lasso; see, e.g., (Breheny & Huang, 2011; 2015; Rakotomamonjy et al., 2019). If additional prior information is available, the works (Deng et al., 2020, May; Gaines et al., 2018; James et al., 2013; Won et al., 2019) propose the constrained Lasso model. The linearly constrained LASSO with nonsmooth nonconvex penalty can be formulated as follows:

$$\min_{x \in \mathbb{X}} \quad \tfrac{1}{2} \|Ax - b\|^2 + \sum_{i=1}^{d} \phi(x_i)$$
$$\text{s.t.} \quad Bx - c = 0 \tag{3}$$

where $x = (x_1, ..., x_d)^\top$, $\mathbb{X} \subset \mathbb{R}^d$, $A \in \mathbb{R}^{n \times d}$ is the design matrix, $b \in \mathbb{R}^n$ is the response vector, $B \in \mathbb{R}^{m \times d}$ and $c \in \mathbb{R}^m$ are given constraints, and $\phi$ is a nonsmooth nonconvex regularizer such as MCP (Zhang et al., 2010) or SCAD (Fan & Li, 2001) penalty.

There are plenty of other applications that give rise to (P), e.g., the robust M-estimators, distributed learning, etc. Due to the limitation of space, we will not expand them in details here.

**Related works.** Linearly constrained nonconvex optimization. Perhaps the most widely utilized class of algorithms for solving constrained optimization problems are the Primal-dual methods. Previous works on single-loop primal-dual methods mainly consider linearly contained convex optimization problems. The work (Boţ & Nguyen, 2020) considers a nonconvex instance with a special linear constraint $Ax - z = 0$, where both $x$ and $z$ are decision variables and the matrix $A$ is assumed to be surjective. Under these assumptions, convergence results were established for a proximal alternating direction method of multipliers (ADMM). The recent works (Zhang & Luo, 2020a;b) provide convergence results of ADMM using gradient-based updates for linearly constrained smooth nonconvex optimization problems with a *general* linear constraint $Ax = b$, i.e., problem (P) with $g \equiv 0$. The main insights of their results are a construction of an proximally regularized auxiliary problem that dates back to (Bertsekas, 1979) and a properly constructed Lyapunov function. Then, they established descent property on the Lyapunov function based on the dual error bound condition (see also (Hong & Luo, 2017)), which leads to iteration complexity results. The method introduced in (Zhu et al., 2020) can be used to solve problem (P) with specific linear constraints. This method is based on the auxiliary problem principle of augmented Lagrangian (APP-AL) method (Cohen & Zhu, 1984; Zhao & Zhu, 2019; 2021), which can be seen as a forward-backward splitting method and applies a Jacobian updating strategy.[1] However, both ADMM-type and APP-AL-type methods compute the full gradient of primal variables and update all primal variables at each iteration. Therefore, the computation complexity of one iteration of these two types of methods are expensive for large-scale problems (See Remark 2.1).

Randomized coordinate methods for unconstrained optimization. In the past decade, big data applications are ubiquitous in machine learning. The formulated optimization problems often involve very large datasets, and hence computing the function value or the gradient can be very expensive. These

---

[1] ADMM applies a Gauss-Seidel like minimization strategy, which is related to the Douglas-Rachford splitting method.

observations motivate the study of the randomized coordinate methods. The researches on this type of methods can be traced back to (Nesterov, 2012; 2014). In (Nesterov, 2012), Nesterov studied the iteration complexity of the randomized coordinate gradient (RCG) descent method for smooth convex optimization. Later, in (Nesterov, 2014), the same author analyzed the randomized coordinate subgradient method for a set of piece-wise linear nonsmooth convex optimization problems. The works (Richtárik & Takáč, 2014) and (Lu & Xiao, 2015) extends Nesterov's results to convex additive composite optimization. Iteration complexity and linear convergence of RCG were studied in (Patrascu & Necoara, 2015) and (Karimi et al., 2016) for nonsmooth nonconvex optimization and smooth nonconvex optimization, respectively. Moreover, by employing an asynchronous point to evaluate the gradient in each iteration, The works (Liu & Wright, 2015) and (Liu et al., 2014) establish the iteration complexity results of asynchronous RCG for convex smooth optimization and additive composite optimization, respectively.

A Few results on randomized coordinate methods for linearly constrained optimization. Though the randomized coordinate-type methods for unconstrained optimization are extensively studied, there are only a few results for this type of methods concerning coupled linearly constrained optimization problems. The works (Gao et al., 2019; Wang et al., 2014) study the randomized primal-dual coordinate (RPDC) method and provides iteration complexity results for linearly constrained convex optimization. Better complexity results of RPDC were obtained for linearly constrained strongly convex optimization (Xu & Zhang, 2018). The recent works (Zhu & Zhao, 2020; Latafat et al., 2019; Fercoq & Bianchi, 2019; Alacaoglu et al., 2020) provide almost sure (i.e., with probability 1) asymptotic convergence results of RPDC-type methods for linearly constrained convex optimization. It is worth emphasizing that the works (Latafat et al., 2019; Fercoq & Bianchi, 2019; Alacaoglu et al., 2020) establish sequential convergence results (i.e., the convergence of the whole sequence of iterates).

However, the above mentioned works only considers either ADMM-type and APP-AL-type methods for linearly constrained smooth nonconvex optimization problem or the randomized coordinate methods for linearly constrained convex optimization. To the best of our knowledge, *no* previous result concerns convergence and iterate complexity of the randomized coordinate methods for *linearly constrained nonsmooth nonconvex optimization*, which will be the main focus of this paper.

**Main contributions.** In this paper, we aim to solve the large-scale linearly constrained nonsmooth nonconvex optimization problem (P). To tackle it, we propose the Nonconvex Randomized Primal-Dual Coordinate (N-RPDC) method based on an auxiliary problem (see Section 2). Relying on the mild local uniform metric subregularity property (see Section 3.3), we provide two types of convergence results for N-RPDC (see Section 4). We first show that any cluster point of the sequence of iterates generated by N-RPDC is almost surely (i.e., with probability 1) a stationary point of (P). In addition, we also provide an almost sure asymptotic convergence rate of $O(1/\sqrt{k})$, where $k$ represents iteration number. Next, we establish the expected iteration complexity of N-RPDC. Namely, N-RPDC needs at most $O(\varepsilon^{-2})$ number of iterations in order to drive a surrogate stationarity measure below $\varepsilon$ in expectation.

The fundamental aspect to our convergence analysis is a *surrogate stationarity measure* we discovered for analyzing N-RPDC (see Section 3.1), which is one of the main contributions of this work. The standard stationarity measure (i.e., checking KKT conditions) is not directly applicable here due to the stochastic nature of N-RPDC. Instead, we define the notion of reference point and use its distance to the iterate generated by N-RPDC as a surrogate stationarity measure. Then, such a surrogate stationarity measure is clarified by showing that it is an upper bound of the standard stationarity measure *at the reference point* up to a numerical constant (see Proposition 3.1 and Remark 3.1).

There are also some other interesting techniques that we utilized for establishing the above convergence results. For example, the utilization of the local uniform metric subregularity is crucial in order to deal with the nonsmooth nonconvex term $g$ in problem (P), as it is far from obvious how to prove the previously used error bound condition if $g$ is not null (see Section 3.3).

**Notations.** We use $\langle \cdot, \cdot \rangle$ and $\| \cdot \|$ to denote the Euclidean inner product and the Euclidean norm, respectively. For a matrix $A$, its minimum eigenvalue is denoted by $\lambda_{\min}(A)$. The spectral norm of a matrix $A$ is denoted by $\|A\|$. Let $\mathbb{S}$ be a subset of $\mathbb{R}^d$. We use $\mathrm{proj}_{\mathbb{S}}(z)$ to denote the orthogonal projector onto $\mathbb{S}$ and $\mathrm{dist}(z, \mathbb{S}) := \inf_{x \in \mathbb{S}} \|x - z\|$ to denote the distance between $x$ and $\mathbb{S}$. When

$\mathbb{S} = \emptyset$, we set $\text{dist}(z, \mathbb{S}) = +\infty$. $\mathcal{I}_{\mathbb{S}}(z) = \left\{ \begin{array}{ll} 0, & z \in \mathbb{S} \\ +\infty, & z \notin \mathbb{S} \end{array} \right.$ represents the indicator function of the set $\mathbb{S}$. If $\mathbb{S}$ is a convex set, then the limiting normal cone to $\mathbb{S}$ is defined as $\mathcal{N}_{\mathbb{S}}(z) = \{\xi : \langle \xi, \zeta - z \rangle \leq 0, \forall \zeta \in \mathbb{S}\}$.

## 2 AUXILIARY PROBLEM AND PROPOSED ALGORITHM

In this section, we will define an auxiliary problem of (P). An important feature of the auxiliary problem is that it has exactly the same set of KKT points to the original problem (P). Furthermore, the auxiliary problem has a better geometric structure, which makes it easier to solve. Finally, we will propose a primal-dual coordinate method based on the auxiliary problem.

Let us first present some preliminaries on the subdifferential and KKT conditions. Due to the fact that problem (P) is highly nonsmooth and nonconvex, our general purpose is to design an efficient algorithm for finding a stationary point rather than a globally optimal solution. Hence, we will define certain suitable stationarity measures in this subsection.

Recall that the function $g$ in problem (P) is $\tau$-weakly convex. By (Vial, 1983, Proposition 4.6), we have

$$\partial g(x) = \partial h(x) - \tau x, \tag{4}$$

where $h$ is the associated convex function such that $g(x) = h(x) - \frac{\tau}{2}\|x\|^2$ (see (Vial, 1983, Proposition 4.3) for the guarantee of the existence of such a function $h$) and $\partial h(x)$ is the usual convex subdifferential. Thus, the subdifferential of a weakly convex function is always well defined.

The *Lagrangian* of problem (P) can be written as

$$\mathcal{L}(x, p) = f(x) + g(x) + \langle p, Ax - b \rangle,$$

where $p$ is the dual variable. Then, we have the following *KKT conditions* for problem (P):

$$0 \in \partial\mathcal{L}(x, p) := \left( \begin{array}{c} \nabla f(x) + \partial g(x) + \mathcal{N}_{\mathbb{X}}(x) + A^\top p \\ Ax - b \end{array} \right). \tag{5}$$

A feasible point $(x, p) \in \mathbb{X} \times \mathbb{R}^n$ is called a *stationary point* of (P) if it satisfies the above KKT conditions, i.e., it satisfies $\text{dist}(0, \partial\mathcal{L}(x, p)) = 0$.

### 2.1 THE AUXILIARY PROBLEM

Thanks to the construction used in (Bertsekas, 1979), we introduce an auxiliary problem to (P) in the following:

$$\min_{x \in \mathbb{X}, z \in \mathbb{R}^d} \quad F^+(x, z) = f(x) + g(x) + \frac{\sigma}{2}\|x - z\|^2 \tag{P$^+$}$$
$$\text{s.t} \quad Ax - b = 0,$$

where $\sigma > L_f + \rho_g$ is a regularization parameter.

We now characterize the set of stationary points of (P$^+$). Towards that end, let $w = (x, z, p) \in \mathbb{X} \times \mathbb{R}^d \times \mathbb{R}^n$. The Lagrangian of (P$^+$) is given by

$$\mathcal{L}^+(w) = \mathcal{L}^+(x, z, p) = f(x) + g(x) + \frac{\sigma}{2}\|x - z\|^2 + \langle p, Ax - b \rangle.$$

Thus, we have the following KKT conditions for problem (P$^+$):

$$0 \in \partial\mathcal{L}^+(w) = \partial\mathcal{L}^+(x, z, p) := \left( \begin{array}{c} \nabla f(x) + \partial g(x) + \sigma(x - z) + \mathcal{N}_{\mathbb{X}}(x) + A^\top p \\ \sigma(z - x) \\ Ax - b \end{array} \right). \tag{6}$$

A point $w = (x, z, p) \in \mathbb{X} \times \mathbb{R}^d \times \mathbb{R}^n$ satisfying the above KKT conditions is called a stationary point of (P$^+$). One crucial feature of the auxiliary problem is that it has exactly the same set of stationary points as that of the original problem (P), which is presented in the following lemma.

**Lemma 2.1** (relation between stationary points of (P) and (P$^+$)). *The following two statements are equivalent: (a) $(x^*, p^*)$ is a stationary point of problem (P); and (b) $(x^*, x^*, p^*)$ is a stationary point of problem (P$^+$).*

---

**Algorithm 1** N-RPDC: Nonconvex Randomized Primal-Dual Coordinate Method

---

**Initialization:** set $x^0 \in \mathbb{X}$, $z^0 \in \mathbb{R}^d$, and $p^0 \in \text{Null}(A^\top)$; Step sizes $\eta$, $\alpha_x$, and $\alpha_z$.

1: **for** $k = 0, 1, \ldots$ **do**
2:     Update $p$ as $p^{k+1} = p^k + \eta(Ax^k - b)$.
3:     Choose $i(k)$ from $\{1, ..., N\}$ uniformly at random;
4:     Update $x$ and $z$ through coordinate steps:

$$x^{k+1} = \arg\min_{x \in \mathbb{X}} \left\langle \nabla_{i(k)} f(x^k) + \sigma(x^k - z^k)_{i(k)}, x_{i(k)} \right\rangle + g_{i(k)}(x_{i(k)})$$

$$+ \left\langle p^{k+1} + \gamma(Ax^k - b), A_{i(k)} x_{i(k)} \right\rangle + \frac{D(x, x^k)}{\alpha_x},$$

$$z^{k+1} = \arg\min_{z \in \mathbb{R}^d} \left\langle \sigma(z^k - x^k)_{i(k)}, z_{i(k)} \right\rangle + \frac{1}{2\alpha_z} \|z - z^k\|^2.$$

5: **end for**

---

*Proof.* We first prove that part (a) implies part (b). Since $(x^*, p^*)$ is a stationary point of (P), we have $0 \in \partial \mathcal{L}(x^*, p^*)$. Upon plugging $z = x^*$ in the $\mathcal{L}^+(x^*, z, p^*)$ shows that $0 \in \partial \mathcal{L}^+(x^*, x^*, p^*)$. The inverse direction is proved by a similar argument. □

The auxiliary problem can be viewed as a quadratically regularized version of the original problem (P) with an additional variable $z$. Recall that $f$ has $L_f$-Lipschitz gradient and $g$ is $\rho_g$-weakly convex. We have the following benign properties of the auxiliary problem (P$^+$), whose derivations can be found in Appendix A.2:

(1) $f(x) + \frac{\sigma}{2}\|x - z\|^2$ is continuously differentiable in $(x, z)$ with Lipschitz continuous gradient;

(2) $F^+(x, z) = f(x) + g(x) + \frac{\sigma}{2}\|x - z\|^2$ is weakly convex in $(x, z)$ with parameter $L_f + \rho_g$;

(3) $F^+(x, z)$ is bi-strongly convex, i.e., $F^+(x, z)$ is strongly convex in $x$ for every fixed $z$ and $F^+(x, z)$ is strongly convex in $z$ for every fixed $x$.

## 2.2 Nonconvex Randomized Primal-Dual Coordinate Method

As we established in the last subsection, problem (P$^+$) has the same set of stationary points as that of the original problem (P) and the former has a series of benign properties. These observations motivate us to design algorithm for solving the original problem by targeting on (P$^+$). Similar to standard primal-dual methods, our algorithm for solving problem (P$^+$) builds on the following *augmented Lagrangian* of (P$^+$):

$$\mathcal{L}_\gamma^+(x, z, p) = f(x) + g(x) + \frac{\sigma}{2}\|x - z\|^2 + \langle p, Ax - b \rangle + \frac{\gamma}{2}\|Ax - b\|^2. \tag{7}$$

With the augmented Lagrangian, our algorithm is designed by performing an APP-AL update over $(x, z, p)$ at each iteration. Furthermore, the updates for variables $x$ and $z$ are achieved through *randomized coordinate* steps. We depict our algorithmic procedures—i.e., the N-RPDC method—in Algorithm 1.

The proximal term $D(x, x^k)$ used in N-RPDC is the so-called *Bregman distance function*. Let $\beta_K$ be the strong convexity parameter of the Bregman distance function $D(x, x^k)$. It is safe for the readers to think $D(x, x^k) = \frac{1}{2}\|x - x^k\|^2$ for now. We put the more general choices of $D(x, x^k)$ in Appendix A.3.

It is worth mentioning that the subproblem for updating $x^{k+1}$ in N-RPDC has a closed form solution once the Bregman distance function $D(x, x^k) = \frac{1}{2}\|x - x^k\|^2$ and $g_i(x_i)$ is SCAD, MCP, quadratic, or $\ell_\nu$-norms with $\nu \in \{1, 2, \infty\}$.

**Remark 2.1** (computational complexity)**.** It is claimed in (Nesterov, 2012; Wright, 2015) that a coordinate method can have a much lower computational complexity than its full counterpart. Let us take the second motivating application (i.e., nonsmooth nonconvex constrained Lasso, problem (3) in Application 2) in Section 1 as an illustrative example to show that the computational complexity of one iteration of N-RPDC with $N = d$ can be much cheaper than the full primal-dual counterpart (ADMM-type or APP-type method). Note that in this case, each $\mathbb{X}_i$ is a box constraint on the coordi-

nate $x_i$. Suppose we set $D(x, x^k) = \frac{1}{2}\|x - x^k\|^2$. Then, the update of $w^{k+1} = (x^{k+1}, z^{k+1}, p^{k+1})$ consists of: 1) $p^{k+1} = p^k + \eta s^k$. 2) $x_{i(k)}^{k+1} = \text{prox}_{\alpha_x, \mathcal{I}_{\mathbb{X}_{i(k)}} + \phi}\left(x_{i(k)}^k - \alpha_x((A_{i(k)})^\top r^k + \sigma(x^k - z^k)_{i(k)} + (B_{i(k)})^\top (p^{k+1} + \gamma s^k))\right)$ and $x_{j \neq i(k)}^{k+1} = x_{j \neq i(k)}^k$, where $\mathcal{I}_{\mathbb{X}_{i(k)}}$ is the indicator function of $\mathbb{X}_{i(k)}$. 3) $z_{i(k)}^{k+1} = z_{i(k)}^k - \alpha_z \sigma(z^k - x^k)_{i(k)}$ with $z_{j \neq i(k)}^{k+1} = z_{j \neq i(k)}^k$. We also update the residual vectors as $r^{k+1} = r^k + A_{i(k)}(x_{i(k)}^{k+1} - x_{i(k)}^k)$ and $s^{k+1} = s^k + B_{i(k)}(x_{i(k)}^{k+1} - x_{i(k)}^k)$. If we precompute $r^0 = Ax^0 - b$ and $s^0 = Bx^0 - c$, and the proximal mapping $\text{prox}_{\alpha_x, \mathcal{I}_{\mathbb{X}_{i(k)}} + \phi}$ admits a closed form solution (which is the case when $\phi$ is the SCAD or MCP penalty), then the above updates have a total complexity $O(\max\{m, n\})$. Therefore, the complexity of N-RPDC is $d\times$ cheaper than its full counterpart (which has a computational complexity of $O(d \max\{m, n\})$ in one iteration).

## 3 STATIONARITY MEASURE, LYAPUNOV FUNCTION, AND UNIFORM METRIC SUBREGULARITY

In this section, we will establish a series of important results, which serve as the foundations for our later convergence analysis.

### 3.1 THE STATIONARITY MEASURE

Consider the current iterate $w^k$. If $w^{k+1}$ is updated by the *full* primal-dual counterpart to N-RPDC (i.e., $N = 1$ in Algorithm 1), then the step length $\|w^k - w^{k+1}\|$ is a natural stationarity measure due to the facts that $\text{dist}(0, \partial\mathcal{L}^+(w^{k+1})) = O(\|w^k - w^{k+1}\|)$ with some step sizes hidden in the big-O and $\lim_{k \to +\infty} \|w^k - w^{k+1}\| = 0$ can be established by an easy argument based on an appropriate Lyapunov function. However, if $w^{k+1}$ is updated by our N-RPDC with $N \neq 1$, which is a stochastic algorithm that only updates *part* of $w^k$ to $w^{k+1}$, it is no longer obvious to show $\lim_{k \to +\infty} \|w^k - w^{k+1}\| = 0$. Therefore, due to the *stochastic nature* of N-RPDC, the step length $\|w^k - w^{k+1}\|$ cannot play the role of stationarity measure. This observation motivates us to work with a surrogate stationarity measure. Towards that end, for a given point $w = (x, z, p)$, we define the reference point of $w$ as $T(w) := (T_x(w), T_z(w), T_p(w))$ and

$$
\begin{cases}
T_p(w) = p + \eta(Ax - b); \\
T_x(w) = \arg\min_{y \in \mathbb{X}} \langle \nabla f(x) + \sigma(x - z), y \rangle + g(y) + \langle T_p(w) + \gamma(Ax - b), Ay \rangle + \dfrac{D(y, x)}{\alpha_x}; \\
T_z(w) = \arg\min_{y \in \mathbb{R}^d} \langle \sigma(z - x), y \rangle + \dfrac{1}{2\alpha_z}\|y - z\|^2.
\end{cases}
\tag{8}
$$

One possible way to understanding the reference point is to let $w^k = (x^k, z^k, p^k)$, then $T(w^k)$ is nothing else other than the next iterate $w^{k+1}$ if the full primal-dual method is utilized, i.e., $N = 1$ in N-RPDC. However, it is worthing mentioning in the case where $N > 1$, we do *not* need to compute the reference point $T(w^k)$ explicitly. Instead, N-RPDC only use a block coordinate for updating.

The next proposition provides us a valid stationary measure, which is crucial for our later convergence analysis.

**Proposition 3.1** (surrogate stationarity measure). *For all $w = (x, z, p) \in \mathbb{X} \times \mathbb{R}^d \times \mathbb{R}^n$, we have*

$$
\boxed{\text{dist}\left(0, \partial\mathcal{L}^+(T(w))\right) \leq c\|w - T(w)\| =: \Phi(w).}
$$

*Here, $T(w)$ is the reference point of $w$ defined in (8) and $c := \sqrt{3} \max\left\{\frac{L_K}{\alpha_x} + L_f + 2\sigma + \|A\|, \frac{1}{\alpha_z} + 2\sigma, \frac{\gamma\|A\|+1}{\eta}\right\}$ is a positive constant.*

**Remark 3.1** (interpretation of the surrogate stationarity measure). The mapping $\Phi(w)$ will serve as the surrogate stationarity measure for analyzing the convergence of N-RPDC. Particularly, it is clear that $\text{dist}\left(0, \partial\mathcal{L}_\gamma^+(T(w))\right) = 0$ when $\Phi(w) = 0$. In order to characterize the iteration complexity of N-RPDC, we focusing on achieving $\Phi(w) \leq \varepsilon$ since it implies that $w$ *is $\varepsilon$-close to the reference point $T(w)$ which is an $\varepsilon$-stationary point* (here, $T(w)$ is defined to be $\varepsilon$-stationary since

dist $(0, \partial \mathcal{L}^+(T(w))) \leq \varepsilon)$. These interpretations clarify our surrogate stationarity measure. Note that similar notion of surrogate stationarity measure dates back to Ekeland's variational principle (Ekeland, 1974) and also appeared in recent advances of the analysis of the subgradient-type methods; see, e.g., (Davis & Drusvyatskiy, 2019; Li et al., 2021). It is interesting to note that the main difference between our approach and the existing ones lies in the choice of the reference point. Let us take the approach used in (Davis & Drusvyatskiy, 2019) for analyzing subgradient-type method as an example. The authors define the proximal mapping of the objective function as the reference point, while we utilize the update of the full primal-dual method (i.e., $N = 1$ in Algorithm 1) as the reference point. Such a main difference comes from different sources of difficulties. The main difficulty for the analysis of subgradient-type method in (Davis & Drusvyatskiy, 2019) comes from the fact that this algorithm is intrinsically not a descent method on the objective function, while our main difficulty is due to the stochastic nature of N-RPDC.

## 3.2 THE LYAPUNOV FUNCTION AND ONE-STEP ANALYSIS

Recall that the function $F^+$ defined in (P$^+$) is strongly convex in $x$ for every fixed $z$ (see our analysis in Section 2.1). Let us define

$$\nu(z) = \min_{x \in \mathbb{X}, \ Ax-b=0} F^+(x,z) \quad \text{and} \quad x(z) = \arg\min_{x \in \mathbb{X}, \ Ax-b=0} F^+(x,z). \tag{9}$$

In addition, we also define

$$\psi_\gamma(z,p) = \min_{x \in \mathbb{X}} \mathcal{L}_\gamma^+(x,z,p) \quad \text{and} \quad x(z,p) = \arg\min_{x \in \mathbb{X}} \mathcal{L}_\gamma^+(x,z,p), \tag{10}$$

where $\mathcal{L}_\gamma^+$ is defined in (7).

Next, mimicking the construction of the Lyapunov function used in (Zhang & Luo, 2020b), we define

$$\Lambda(w) = \mathcal{L}_\gamma^+(w) + 2\left(\nu(z) - \psi_\gamma(z,p)\right) \tag{11}$$

as the Lyapunov function for our convergence analysis. The first part of $\Lambda(\cdot)$ is augmented Lagrangian $\mathcal{L}_\gamma^+(w)$ of (P$^+$). The second part is dual gap of (P$^+$) with fixed $z$. Additionally, by the definition of $\psi_\gamma(z,p)$, we have that $\mathcal{L}_\gamma^+(w) \geq \psi_\gamma(z,p)$. By the strong duality, we have $\nu(z) = \max_p \psi_\gamma(z,p)$. Therefore, we have the fact that $\Lambda(w) \geq \nu(z) \geq F^*$.

Standard analysis for primal-dual methods develops certain descent property on the Lyapunov function. Before that, we first present a preliminary one-step analysis for N-RPDC. Note that indices $i(k)$, $k = 0, 1, 2, \ldots$ in N-RPDC are random variables. Thus, it generates a random output. We use

$$\mathcal{F}_k := \{i(0), i(1), \ldots, i(k)\}$$

to denote the filtration generated by the random variables $i(0), i(1), \ldots, i(k)$. Clearly, we have $\mathcal{F}_k \subset \mathcal{F}_{k+1}$. We use $\mathbb{E}_{i(k)}$ and $\mathbb{E}_{\mathcal{F}_k}$ to denote the expectations taken over the random variable $i(k)$ and the filtration $\mathcal{F}_k$, respectively. The following lemma provides estimations for $\nu(z) - \psi_\gamma(z,p)$ and $\mathcal{L}_\gamma^+(w)$ after one-step execution of N-RPDC.

**Lemma 3.1** (one-step analysis of N-RPDC). *Suppose* $0 < \alpha_x \leq \beta_K / \left[L_f + 2\sigma + \gamma\|A\|^2 + 5\right]$ *and* $0 < \eta < 1 / \left[2N\|A\|^2 \left(\frac{L_f + \sigma + \gamma\|A\|^2 + \frac{L_K}{\alpha_x}}{\sigma - L_f - \rho_g} + 1\right)^2\right]$. *For all $k \geq 0$, we have*

$$\begin{aligned}
\Lambda(w^k) &- \mathbb{E}_{i(k)} \Lambda(w^{k+1}) \\
&\geq \frac{2}{N}\|x^k - T_x(w^k)\|^2 + \frac{1}{N}\left[\frac{1}{\alpha_z} - 2\sigma - \frac{\sigma}{\lambda} - \frac{3\sigma^2}{\sigma - L_f - \rho_g}\right]\|z^k - T_z(w^k)\|^2 \quad (12) \\
&+ \eta\|Ax(z^k, T_p(w^k)) - b\|^2 - \sigma\lambda\|x(z^k, T_p(w^k)) - x(z^k)\|^2.
\end{aligned}$$

*where $\sigma$ is defined in (P$^+$) and $\lambda$ is any positive constant.*

Toward establishing certain decent property on the Lyapunov function in terms of the stationarity measure $\Phi(w^k)$, we have to connect all the terms on the right-hand side of the inequality of Lemma 3.1 to $\Phi^2(w^k) = \|w^k - T(w^k)\|^2$. Thus, it remains to upper bound $\|x(z^k, T_p(w^k)) - x(z^k)\|$ by $\|Ax(z^k, T_p(w^k)) - b\|$, which needs the so-called uniform metric subregularity property.

### 3.3 Uniform Metric Subregularity and its Consequence

Let us define the parameter set-valued mapping

$$\partial \mathcal{L}_\gamma^z(x, p) = \begin{pmatrix} \nabla f(x) + \partial g(x) + \sigma(x - z) + \mathcal{N}_{\mathbb{X}}(x) + A^\top(p + \gamma(Ax - b)) \\ Ax - b \end{pmatrix} \quad (13)$$

as the subdifferential of $L_\gamma^+$ with respect to $(x, p)$ with fixed $z$. Clearly, we have $\partial \mathcal{L}_\gamma^z(x(z, p), p) = \begin{pmatrix} 0 \\ Ax(z, p) - b \end{pmatrix}$ due to the optimality of $x(z, p)$ in (10). To proceed, we introduce the notion of local uniform metric subregularity property for the parameter set-valued mapping.

**Definition 3.1** (local uniform metric subregularity, cf. (Kruger & Duy Cuong, 2021)). *Let $\mathcal{H}_z(u)$ : $\mathbb{U} \rightrightarrows \mathbb{V}$ be a set-valued mapping. We call that $\mathcal{H}_z(u)$ satisfies the locally metric subregularity property with parameter $(\delta, \kappa)$ uniformly over all $z \in \mathbb{Z}$ if*

$$\mathrm{dist}\left(u, \mathcal{H}_z^{-1}(0)\right) \leq \kappa \cdot \mathrm{dist}\left(0, \mathcal{H}_z(u)\right), \quad \forall z \in \mathbb{Z}, \quad (14)$$

*for any $u \in \{u' \in \mathbb{U} : \mathrm{dist}(0, \mathcal{H}_z(u')) \leq \delta\}$, where $\delta, \kappa > 0$ are uniform constants.*

This local error bound condition gives the following result.

**Lemma 3.2.** *Suppose the set-valued mapping $\partial \mathcal{L}_\gamma^z(x, p)$ defined in (13) satisfies the local uniform metric subregularity property, then there is $\kappa > 0$ (independent of z) such that*

$$\|x(z, p) - x(z)\| \leq \kappa \|Ax(z, p) - b\|, \quad \text{whenever } \|Ax(z, p) - b\| \leq \delta.$$

Equipped with this lemma, we are able to upper bound $\|x(z^k, T_p(w^k)) - x(z^k)\|$ by $\|Ax(z^k, T_p(w^k)) - b\|$ *locally* in order to establish the descent property on the Lyapunov function. We will discuss this descent property in the next section. Now, let us provide the conditions on problem (P) in order to ensure the local uniform metric subregularity assumed in Lemma 3.2.

**Lemma 3.3** (sufficient condition for local uniform metric subregularity). *Consider problem (P). Suppose that $\nabla f(x)$ is piecewise affine and $\partial g(x)$ is polyhedral multifunction and $\mathbb{X}$ is a polyhedral set, then $\partial \mathcal{L}_\gamma^z(x, p)$ defined in (13) satisfies the local uniform metric subregularity property.*

**Remark 3.2** (generality of local uniform metric subregularity). This sufficient condition ensures that $\partial \mathcal{L}_\gamma^z(x, p)$ must satisfy the local uniform metric subregularity property whenever $f$ is quadratic (can be nonconvex, i.e., its Hessian is not required to be positive semidefinite), $g$ can be written as $\|\cdot\|_1 - \frac{\rho_g}{2}\|\cdot\|^2$, $\ell_2$-norm, $\ell_\infty$-norm structure, and $\mathbb{X}$ is a polyhedron, which covers all the two motivating applications listed in Section 1.

**Remark 3.3** (comparison to the error bound condition utilized in (Zhang & Luo, 2020b)). Recall that the work (Zhang & Luo, 2020b) considers problem (P) with $g \equiv 0$. They use the so-called dual error bound condition (see (Hong & Luo, 2017)) instead of the local uniform metric subregularity property. However, we argue that it is far form obvious how to extend their technique to cover our nonsmooth nonconvex case, as it is not clear whether one can establish the dual error bound condition if $g$ is not null.

**Remark 3.4** (further comments on the surrogate stationarity measure $\Phi(w)$ defined in Proposition 3.1). It is interesting that the sufficient condition in Lemma 3.3 also ensures that $\partial \mathcal{L}^+(w)$ satisfies the local metric subregularity when $\Phi(w) \leq \varepsilon$ with sufficient small $\varepsilon$ (Robinson, 1981), i.e., $\mathrm{dist}(w, \overline{\mathbb{W}}) \leq \kappa' \mathrm{dist}(0, \partial \mathcal{L}^+(w))$ for some $\kappa' > 0$, where $\overline{\mathbb{W}}$ is the set of stationary points of (P$^+$). Then, we have $\mathrm{dist}(w, \overline{\mathbb{W}}) \leq (1/c + \kappa')\varepsilon$ which follows from $\mathrm{dist}(T(w), \overline{\mathbb{W}}) \leq \kappa'\varepsilon$ (due to $\Phi(w) \leq \varepsilon$) and the triangle inequality.

## 4 Convergence and Iteration Complexity of N-RPDC

Equipped with all the machineries developed in Section 3, we are now ready to establish the convergence properties of N-RPDC.

### 4.1 Descent Property on the Lyapnov Function

Upon invoking Lemma 3.2 in Lemma 3.1, we can derive the descent property in expectation on the Lyapnov Function.

**Lemma 4.1** (expected sufficient decrease for $\Lambda^k$). *Suppose that Assumptions of Lemma 3.1 hold. If the set-valued mapping $\partial \mathcal{L}_\gamma^z(x, p)$ defined in (13) satisfies the local uniform metric subregularity property (see Definition 3.1). Let $\lambda = \min\{\frac{\delta\eta}{2\sigma M^2 + 1}, \frac{\eta}{\sigma\kappa^2 + 1}\}$, the parameter $\alpha_z$ satisfy $0 < \alpha_z < 1/\left[2\sigma + \frac{\sigma}{\lambda} + \frac{3\sigma^2}{\sigma - L_f - \rho_g} + 1\right]$, and $c_1 = \min\left\{1, \frac{N}{2\eta(\sigma\kappa^2 + 2) + 1}\right\}$. Then, we have*

$$\Lambda(w^k) - \mathbb{E}_{i(k)}\Lambda(w^{k+1}) \geq \frac{c_1}{N}\|w^k - T(w^k)\|^2.$$

### 4.2 ALMOST SURE CONVERGENCE RESULTS AND EXPECTED ITERATION COMPLEXITY

In this subsection, we establish the *almost sure* (i.e., with probability 1) convergence results for N-RPDC.

**Theorem 4.1** (almost sure convergence result). *Under the setting of Lemma 4.1, then*

*(a) $\lim_{k \to +\infty} \Phi(w^k) = 0$ almost surely.*

*(b) The sequence $\{w^k\}$ is almost surely bounded.*

*(c) Any cluster point of the sequence $\{w^k\}$ is almost surely a stationary point of (P).*

In addition, we can also provide the almost surely asymptotic rate of convergence in the sense of limit inferior for our N-RPDC method.

**Theorem 4.2** (almost sure $O(1/\sqrt{k})$ convergence rate). *Under the setting of Lemma 4.1, we have*

$$\liminf_{k \to +\infty} \sqrt{k+1} \cdot \Phi(w^k) = 0 \quad \text{almost surely}$$

*i.e., the asymptotic $O(1/\sqrt{k})$ convergence rate holds almost surely.*

The almost sure convergence rate established above is of asymptotic nature, which concerns the behavior of N-RPDC when $k \to \infty$. Next, we also provide the iteration complexity results of N-RPDC in expectation.

**Theorem 4.3** (expected iteration complexity). *Under the setting of Lemma 4.1. The sequence $\{w^k\}$ is generated by N-RPDC. Then*

$$\min_{0 \leq k \leq t} \mathbb{E}_{\mathcal{F}_t} \Phi(w^k) \leq \sqrt{\frac{Nc^2(\Lambda(w^0) - F^*)}{c_1(t+1)}}.$$

*Consequently, to achieve $\mathbb{E}_{\mathcal{F}_t}\Phi(w^{\bar{t}}) \leq \varepsilon$ for some $0 \leq \bar{t} \leq t$, N-RPDC needs at most $t = \frac{Nc^2(\Lambda(w^0) - F^*)}{c_1\varepsilon^2}$ number of iterations.*

**Remark 4.1.** Note that the closely related work (Zhang & Luo, 2020b) provides a similar iteration complexity result (i.e., $O(\varepsilon^2)$) if $g \equiv 0$. By contrast, our results can deal with the additional nonsmooth nonconvex term $g$. In addition, we also provide the almost sure convergence results, complementing the results in (Zhang & Luo, 2020b). Let us also emphasize that the main motivation of our algorithm lies in that N-RPDC is a randomized coordinate method, which is more suitable for modern large-scale problems.

**Remark 4.2.** The recent work (Zhang & Luo, 2020a) establishes a global dual error bound condition, which allows them to avoid the compactness assumption on $\mathbb{X}$. Due to the existence of $g$, we utilize the local uniform metric subregularity rather than this dual error bound condition. To further relax the compactness assumption on $\mathbb{X}$ in problem (P), one possible direction is to establish a global version of the utilized local uniform metric subregularity. We leave this direction as a future work.

## 5 NUMERICAL EXPERIMENTS

Because of the limitation of space, we put the numerical experiments in Appendix A.1. We can observe that N-RPDC with larger $N$ slightly outperforms that with smaller $N$ in terms of convergence speed. We also compared our N-RPDC to the algorithm proposed in (Zhang & Luo, 2020b); see Figure 1. We can observe that N-RPDC slightly outperforms their algorithm. Note that the main motivation of N-RPDC is that it is more suitable for modern large-scale problems due to its reasonably fast convergence speed and low computational complexity at each iteration.

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

## A    APPENDIX

The Appendix is organized as follows. Subsection A.1 discusses implementation details for N-RPDC and presents numerical experiments on non-PSD kernel SVM and sum to zero constrained LASSO problem with SCAD penalty. Some properties of the auxiliary problem (P$^+$) are given in subsection A.2. Some examples of Bregman distance function are given in A.3. Subsection A.4 shows the proof of Proposition 3.1. Subsection A.5 provides the proof of Lemma 3.1. Proof of Lemma 3.2 is given in subsection A.6. Subsection A.7 provides the proof of Lemma 3.3. Subsection A.8 shows the proof of Lemma 4.1. Subsection A.9 provides the proof of Theorem 4.1. Subsection A.10 gives the proof of Theorem 4.2. The proof of Theorem 4.3 is given in subsection A.11.

### A.1    NUMERICAL EXAMPLES

This subsection discusses experiments conducted using MATLAB R2020a on a personal computer with Intel Core i5-6200U CPU (2.40GHz) and 8.00 GB RAM.

#### A.1.1    NON-PSD KERNEL SUPPORT VECTOR MACHINE PROBLEM

Consider the non-PSD kernel support vector machine problem,

$$\min_{x \in [0,c]^d} \quad \frac{1}{2} x^\top Q x - \mathbf{1}_d^\top x \\ \text{s.t.} \quad y^\top x = 0,$$

where $x \in \mathbb{R}^d$ are the decision variables, and $Q \in \mathbb{R}^{d \times d}$ is a matrix, possibly non-PSD. Let $Q = (Q_1^\top, Q_2^\top, \cdots, Q_N^\top)^\top \in \mathbb{R}^{d \times d}$ be an appropriate partition of matrix $Q$ and $Q_i$ be an $d_i \times d$ matrix. $\mathcal{L}_\gamma^+$ of Non-PSD kernel SVM can be written as

$$\mathcal{L}_\gamma^+(w) = \mathcal{L}_\gamma^+(x, z, p) = \frac{1}{2} x^\top Q x - \mathbf{1}_d^\top x + \frac{\sigma}{2} \|x - z\|^2 + \langle p, y^\top x \rangle + \frac{\gamma}{2} \|y^\top x\|^2.$$

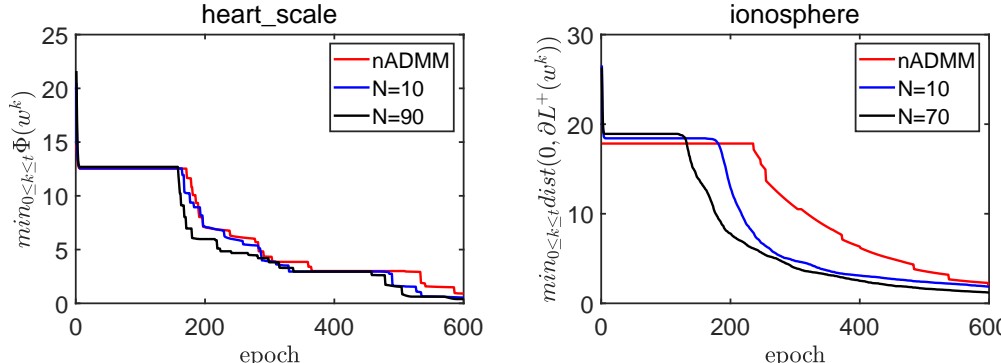

Figure 1: Evolution of $\min_{0 < k \leq t} \Phi(w^k)$ versus epoch counts for several different choices of $N$ (i.e., the number of blocks in N-RPDC), where nADMM refers to the algorithm proposed in (Zhang & Luo, 2020b). Left: dataset `heart_scale`. Right: dataset `ionosphere`. The results are obtained by averaging 50 independent trials.

Thanks of Proposition 3.3, for problem Non-PSD kernel SVM, $\partial \mathcal{L}_\gamma^z$ is locally metric subregular in $(x, z, p)$ uniformly in $z$ at point 0. N-RPDC with $D(x, x^k) = \frac{1}{2}\|x - x^k\|^2$ for non-PSD kernel support vector machine problem is

$$p^{k+1} \leftarrow p^k + \eta y^\top x^k;$$

Choose $i(k)$ from $\{1, 2, \ldots, N\}$ with equal probability

$$x^{k+1} \leftarrow \min_{x \in [0,c]^d} \langle Q_{i(k)} x^k + \sigma(x^k - z^k)_{i(k)} - \mathbf{1}_{d_{i(k)}} + y_{i(k)} \left( p^{k+1} + \gamma y^\top x^k \right), x_{i(k)} \rangle$$

$$+ \frac{1}{2\alpha_x}\|x - x^k\|^2;$$

$$z^{k+1} \leftarrow \min_{z \in \mathbb{R}^d} \langle \sigma(z^k - x^k)_{i(k)}, z_{i(k)} \rangle + \frac{1}{2\alpha_z}\|z - z^k\|^2.$$

Thus, the primal subproblem of N-RPDC has the closed form

$$\begin{cases} x_{i(k)}^{k+1} = \Pi_{[0,c]^{d_{i(k)}}} \left[ x_{i(k)}^k - \alpha_x \left( Q_{i(k)} x^k + \sigma(x^k - z^k)_{i(k)} - \mathbf{1}_{n_{i(k)}} + (p^k + \gamma y^\top x^k) y_{i(k)} \right) \right], \\ x_{j \neq i(k)}^{k+1} = x_{j \neq i(k)}^k; \end{cases}$$

and

$$\begin{cases} z_{i(k)}^{k+1} = z_{i(k)}^k - \alpha_z \sigma(z^k - x^k)_{i(k)}, \\ z_{j \neq i(k)}^{k+1} = z_{j \neq i(k)}^k. \end{cases}$$

We used LIBSVM dataset `heart_scale` and `ionosphere` in the experiment. $Q$ was generated using the sigmoid kernel (Lin & Lin, 2003), and we selected $c = 1$. We compared two algorithms: nonconvex ADMM in (Zhang & Luo, 2020b) (**nADMM**), and **N-RPDC** from this paper on these two datasets.

For N-RPDC algorithm, we partitioned the variables $N = 10, 90$ blocks for `heart_scale` and $N = 10, 70$ blocks for `ionosphere`.

In order to verify the theoretical results of this paper, we compute the sequence of reference point $\{T(w^k)\}$ along with the sequence $\{w^k\}$ generated by N-RPDC. In the real world applications of N-RPDC, the reference point sequence is not necessary to compute. In Figure 1, the left graph show the number of blocks and $\min_{0 \leq k \leq t} \Phi(w^k)$ with respect to epoch count for `heart_scale`; the right graph show the number of blocks and $\min_{0 \leq k \leq t} \Phi(w^k)$ with respect to epoch count for `ionosphere`. For both datasets, $\min_{0 \leq k \leq t} \Phi(w^k)$ of each epoch of N-RPDC is the average value of 50 times result. Moreover, we can draw the conclusion that N-RPDC is slightly outperforms nADMM and larger $N$ is slightly outperforms small $N$.

A.1.2 SUM TO ZERO CONSTRAINED LASSO PROBLEM WITH SCAD PENALTY

Consider the sum to zero constrained LASSO problem with SCAD penalty:

$$\text{(CLASSO-SCAD)} \quad \min_{x \in [-1,1]^d} \quad \frac{1}{2}\|Ax - b\|^2 + \sum_{i=1}^n \phi(x_i) \ ,$$
$$\text{s.t.} \quad \mathbf{1}_n^\top x = 0;$$

where $x \in \mathbb{R}^d$ is the decision variables and $A \in \mathbb{R}^{n \times d}$ is the regressors matrix. Let $A = (A_1, A_2, \cdots, A_N) \in \mathbb{R}^{n \times d}$ be an appropriate partition of matrix $A$ and $A_i$ be an $n \times d_i$ matrix. $b \in \mathbb{R}^n$ is the response vector. $\mathbf{1}_d$ is an $d$-dimensional vector of 1s. $\phi(\cdot)$ is the SCAD penalty:

$$\phi(|t|) = \begin{cases} |t| & |t| \leq 1 \\ \frac{-t^2 + 2\theta|t| - 1}{2(\theta-1)} & 1 < |t| \leq \theta \ , \quad t \in \mathbb{R} \\ \frac{1+\theta}{2} & |t| > \theta. \end{cases}$$

$\mathcal{L}_\gamma^+$ of (CLASSO-SCAD) can be written as

$$\mathcal{L}_\gamma^+(w) = \mathcal{L}_\gamma^+(x, z, p) = \frac{1}{2}\|Ax - b\|^2 + \sum_{i=1}^n \phi(x_i) + \frac{\sigma}{2}\|x - z\|^2 + \langle p, \mathbf{1}_n^\top x \rangle + \frac{\gamma}{2}\|\mathbf{1}_n^\top x\|^2.$$

Thanks of Proposition 3.3, for problem (CLASSO-SCAD), $\partial \mathcal{L}_\gamma^z$ is locally metric subregular in $(x, z, p)$ uniformly in $z$ at point 0. We take $D(x, x^k) = \frac{1}{2}\|x - x^k\|^2$ and construct N-RPDC algorithm for (CLASSO-SCAD) as following:

$$p^{k+1} \leftarrow p^k + \eta(\mathbf{1}_d^\top x^k),$$
$$\text{Choose } i(k) \text{ from } \{1, 2, \ldots, N\} \text{ with equal probability}$$
$$x^{k+1} \leftarrow \min_{x \in [-1,1]^d} \langle A_{i(k)}^\top(Ax^k - b) + \sigma(x^k - z^k)_{i(k)} + \mathbf{1}_{d_i}(q^k), x_{i(k)} \rangle$$
$$+ \sum_{i=1+\sum_{j=1}^{i(k)-1} d_j}^{\sum_{j=1}^{i(k)} d_j} \phi(x_i) + \frac{1}{2\alpha_x}\|x - x^k\|^2;$$
$$z^{k+1} \leftarrow \min_{z \in \mathbb{R}^d} \langle \sigma(z^k - x^k)_{i(k)}, z_{i(k)} \rangle + \frac{1}{2\alpha_z}\|z - z^k\|^2;$$

where $q^k = p^{k+1} + \gamma(\mathbf{1}_d^\top x^k)$. Thus, the primal subproblem of RPDC has the closed form

$$\begin{cases} x_{i(k)}^{k+1} = \begin{cases} \Pi_{[-1,1]^{d_{i(k)}}}\{sign(\zeta^k) \odot \max(0, |\zeta^k| - \mu \mathbf{1}_{d_{i(k)}})\} & |\zeta^k| \leq 1 + \mu \\ \Pi_{[-1,1]^{d_{i(k)}}}\{\frac{1}{\theta-1-\mu}[(\theta-1)\zeta^k - \mu\theta sign(\zeta^k)]\} & 1 + \mu < |\zeta^k| \leq \theta \\ \Pi_{[-1,1]^{d_{i(k)}}}\{\zeta^k\} & |\zeta^k| > \theta \end{cases} \\ x_{j \neq i(k)}^{k+1} = x_{j \neq i(k)}^k; \end{cases}$$

with $\zeta^k = x_{i(k)}^k - \alpha_x[A_{i(k)}^\top(Ax^k - b) + \sigma(x^k - z^k)_{i(k)} + q^k \mathbf{1}_{n_{i(k)}}]$ and $\mu = \alpha_x$.

$$\begin{cases} z_{i(k)}^{k+1} = z_{i(k)}^k - \alpha_z \sigma(z^k - x^k)_{i(k)}; \\ z_{j \neq i(k)}^{k+1} = z_{j \neq i(k)}^k. \end{cases}$$

In this experiment, the elements of $A \in \mathbb{R}^{n \times d}$ are selected i.i.d. from a Gaussian $\mathcal{N}(0, 1)$ distribution. To construct a sparse true solution $x^* \in \mathbb{R}^d$, given the dimension $d$ and sparsity $s$, we select $s$ entries of $x^*$ at random to be nonzero and $\mathcal{N}(0, 1)$ normally distributed, and set the rest to zero. The measurement vector $b \in \mathbb{R}^n$ is obtained by $b = Ax^* + \delta$ with the elements of the noise vector $\delta \in \mathbb{R}^n$ are i.i.d. $\mathcal{N}(0, 0.001)$. We choose $\theta = 2.3$ with $n = 360, d = 1280, s = 8$ for the first dataset and $n = 720, d = 2560, s = 16$ for the second dataset.

We partitioned the variables $N = 10, 40, 80$ blocks. Thus $d_i = 128, 32, 16$ for the first dataset and $d_i = 256, 64, 32$ for the second dataset.

In order to verify the theoretical results of this paper, we compute the sequence of reference point $\{T(w^k)\}$ along with the sequence $\{w^k\}$ generated by N-RPDC. In the real world applications of N-RPDC, the reference point sequence is not necessary to compute. In Figure 2, the left graph

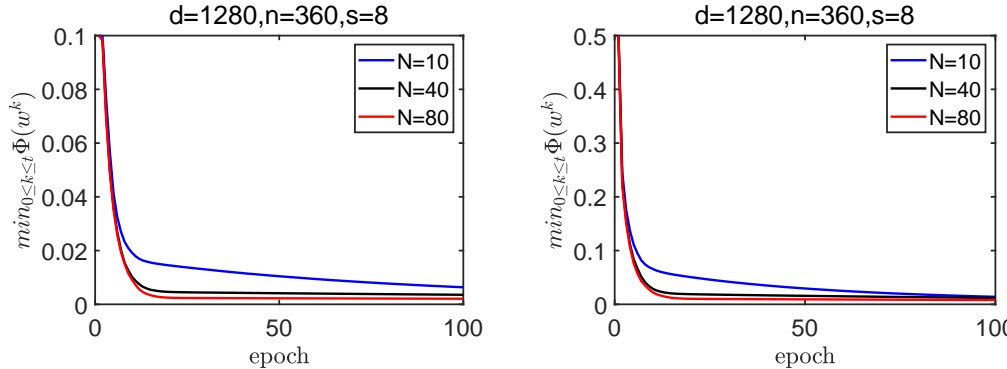

Figure 2: Evolution of $\min_{0 \leq k \leq t} \Phi(w^k)$ versus epoch counts for several different choices of $N$ (i.e., the number of blocks in N-RPDC) on synthetic dataset. Left: $n = 360, d = 1280, s = 8$. Right: $n = 720, d = 2560, s = 16$. The results are obtained by averaging 50 independent trials.

show the number of blocks and $\min_{0 \leq k \leq t} \Phi(w^k)$ with respect to epoch count for the first dataset; the right graph show the number of blocks and $\min_{0 \leq k \leq t} \Phi(w^k)$ with respect to epoch count for the second dataset. For both datasets, $\min_{0 \leq k \leq t} \Phi(w^k)$ for each epoch of N-RPDC is the average value of $50$ times result. Additionally, we can draw the conclusion that larger $N$ is slightly outperforms small $N$.

### A.2 Some properties of the auxiliary problem $P^+$

The auxiliary problem can be viewed as a quadratically regularized version of the original problem (P) with an additional variable $z$. Recall that $f$ has $L_f$-Lipschitz gradient and $g$ is $\rho_g$-weakly convex. We have the following benign properties of the auxiliary problem (P$^+$):

(1) $f(x) + \frac{\sigma}{2}\|x - z\|^2$ is continuously differentiable in $(x, z)$ with Lipschitz continuous gradient;

(2) $F^+(x, z) = f(x) + g(x) + \frac{\sigma}{2}\|x - z\|^2$ is weakly convex in $(x, z)$ with parameter $L_f + \rho_g$;

(3) $F^+(x, z)$ is bi-strongly convex, i.e., $F^+(x, z)$ is strongly convex in $x$ for every fixed $z$ and $F^+(x, z)$ is strongly convex in $z$ for every fixed $x$.

The first property is trivially true. To see the second property, we note that the $L_f$-Lipschitz gradient property of $f$ implies that $f(x) \geq f(y) + \langle s, x - y \rangle - \frac{L_f}{2}\|x - y\|^2, \forall x, y \in \mathbb{R}^d$ (Nesterov, 2003, Lemma 1.2.3). This, together with (Vial, 1983, Proposition 4.8), establishes that $f$ is $L_f$-weakly convex[2]. By (Vial, 1983, Proposition 4.1), we have that $f + g$ is weakly convex with parameter $L_f + \rho_g$. We have shown the second property y recognizing that $(x, z) \mapsto \frac{\sigma}{2}\|x - z\|^2$ is a convex mapping. The third property is a direct consequence of the definition of weak convexity, which yields that $f(x) + g(x) + \frac{\sigma}{2}\|x - z\|^2$ is $\sigma - (L_f + \rho_g)$ strongly convex in $x$.

### A.3 Some examples of Bregman distance function

The proximal term $D(x, x^k)$ used in N-RPDC is the so-called *Bregman distance function*, which is defined as
$$D(x, x^k) := K(x) - K(x^k) - \langle \nabla K(x^k), x - x^k \rangle.$$

Here, $K : \mathbb{R}^d \to \mathbb{R}$ is the core function, which is $\beta_K$-strongly convex and has $L_K$-Lipschitz continuous gradient on $\mathbb{R}^d$. One popular choice of the core function is $K(x) = \frac{1}{2}\|x\|^2$, which yields $D(x, x^k) = \frac{1}{2}\|x - x^k\|^2$. However, more generally, the core function can also be selected as the $Q$-quadratic proximal regularization term $K(x) = \frac{1}{2}\|x\|_Q^2$, where $Q$ is a positive definite matrix. This choice covers some second-order methods as we can set $Q$ to be the regularized Hessian matrix.

---

[2]Actually, this derivation verifies the fact that any continuously differentiable function with $L$-Lipschitz continuous gradient is $L$-weakly convex.

## A.4 Proof of Proposition 3.1

*Proof.* The optimal condition of

$$T_x(w^k) = \arg\min_{x \in \mathbb{X}} \langle \nabla f(x^k) + \sigma(x^k - z^k), x \rangle + g(x) + \langle T_p(w^k) + \gamma(Ax^k - b), Ax \rangle + \frac{D(x,x^k)}{\alpha_x}$$

and $T_z(w^k) = \arg\min_{z \in \mathbb{R}^d} \langle \sigma(z^k - x^k), z \rangle + \frac{1}{2\alpha_z} \|z - z^k\|^2$ yields that

$$0 \in \nabla f(x^k) + \partial g(T_x(w^k)) + \sigma(x^k - z^k) + \mathcal{N}_{\mathbb{X}}(T_x(w^k)) + A^\top[T_p(w^k) + \gamma(Ax^k - b)]$$
$$+ \frac{1}{\alpha_x}[\nabla K(T_x(w^k)) - \nabla K(x^k)];$$

$$0 = \sigma(z^k - x^k) + \frac{1}{\alpha_z}(T_z(w^k) - z^k).$$

Let

$$\begin{aligned}
\xi_x &= \nabla f(T_x(w^k)) - \nabla f(x^k) + \sigma(T_x(w^k) - T_z(w^k)) - \sigma(x^k - z^k) - \gamma A^\top(Ax^k - b) \\
&\quad - \frac{1}{\alpha_x}[\nabla K(T_x(w^k)) - \nabla K(x^k)] \\
&= \nabla f(T_x(w^k)) - \nabla f(x^k) + \sigma(T_x(w^k) - T_z(w^k)) - \sigma(x^k - z^k) - \frac{\gamma}{\eta}A^\top(T_p(w^k) - p^k) \\
&\quad - \frac{1}{\alpha_x}[\nabla K(T_x(w^k)) - \nabla K(x^k)]; \qquad (15)
\end{aligned}$$

$$\xi_z = \sigma(T_z(w^k) - T_x(w^k)) - \sigma(z^k - x^k) - \frac{1}{\alpha_z}(T_z(w^k) - z^k); \qquad (16)$$

and

$$\xi_p = \frac{1}{\eta}(T_p(w^k) - p^k) + A(T_x(w^k) - x^k). \qquad (17)$$

Then we have that

$$\xi_x \in \nabla f(T_x(w^k)) + \partial g(T_x(w^k)) + \sigma(T_x(w^k) - T_z(w^k)) + \mathcal{N}_{\mathbb{X}}(T_x(w^k)) + A^\top T_p(w^k);$$
$$\xi_z = \sigma(T_z(w^k) - T_x(w^k));$$
$$\xi_p = AT_x(w^k) - b;$$

By the expression (15-17) of $\xi = \begin{pmatrix} \xi_x \\ \xi_z \\ \xi_p \end{pmatrix}$, $\alpha_x$, $\alpha_z$ and $\eta$ are given positive numbers and Assumptions of problem (P), there exists positive number $c \geq \sqrt{3}\max\left\{\frac{L_K}{\alpha_x} + L_f + 2\sigma + \|A\|, \frac{1}{\alpha_z} + 2\sigma, \frac{\gamma\|A\|+1}{\eta}\right\}$ and the following inequality holds

$$\|\xi\| \leq c\|w^k - T(w^k)\|.$$

Combining $\xi \in \partial \mathcal{L}^+(T(w^k))$ and it follow the desired statement. $\qquad \square$

## A.5 Proof of Lemma 3.1

In order to provide the Lemma 3.1, we need the following two propositions as preparation.

**Proposition A.1** (Lipschitz properties of $x(z)$ and $x(z,p)$)**.** *Suppose $\sigma > L_f + \rho_g$. Then*

(i) $\|x(z) - x(z')\| \leq \frac{\sigma}{\sigma - (L_f + \rho_g)}\|z - z'\|$;

(ii) $\|x(z,p) - x(z',p)\| \leq \frac{\sigma}{\sigma - (L_f + \rho_g)}\|z - z'\|$.

*Proof.* (i) Let $\phi(x,z) = f(x) + g(x) + \frac{\sigma}{2}\|x-z\|^2 + \mathcal{I}_{\tilde{\mathbb{X}}}(x)$ with $\tilde{\mathbb{X}} = \{x \in \mathbb{X} | Ax - b = 0\}$. Since $\sigma > L_f + \rho_g$, we have $\phi(x,z)$ is $\sigma - (L_f + \rho_g)$-strongly convex in $x$. Therefore

$$\phi\left(x(z), z'\right) - \phi\left(x(z'), z'\right)$$
$$= \ \left[\phi\left(x(z), z'\right) - \phi\left(x(z), z\right)\right] + \left[\phi\left(x(z), z\right) - \phi\left(x(z'), z\right)\right] + \left[\phi\left(x(z'), z\right) - \phi\left(x(z'), z'\right)\right]$$
$$= \ \frac{\sigma}{2}[\|x(z) - z'\|^2 - \|x(z) - z\|^2] + \left[\phi\left(x(z), z\right) - \phi\left(x(z'), z\right)\right]$$
$$+ \frac{\sigma}{2}[\|x(z') - z\|^2 - \|x(z') - z'\|^2]$$
$$= \ \left[\phi\left(x(z), z\right) - \phi\left(x(z'), z\right)\right] + \sigma\langle z' - z, x(z') - x(z)\rangle$$
$$\leq \ -\frac{\sigma - (L_f + \rho_g)}{2}\|x(z) - x(z')\|^2 + \sigma\langle z' - z, x(z') - x(z)\rangle$$

Again using the strongly convex of $\phi(x,z)$ in $x$, we have that

$$\phi\left(x(z), z'\right) - \phi\left(x(z'), z'\right) \geq \frac{\sigma - (L_f + \rho_g)}{2}\|x(z) - x(z')\|^2.$$

Therefore

$$\|x(z) - x(z')\| \leq \frac{\sigma}{\sigma - (L_f + \rho_g)}\|z - z'\|.$$

(ii) Let $\varphi(x,z,p) = \mathcal{L}_\gamma^+(x,z,p) + \mathcal{I}_{\mathbb{X}}(x)$. Since $\sigma > L_f + \rho_g$, function $\varphi(x,z,p)$ also is $\sigma - (L_f + \rho_g)$-strongly convex in $x$. Therefore

$$\varphi\left(x(z,p), z', p\right) - \varphi\left(x(z',p), z', p\right)$$
$$= \ \left[\varphi\left(x(z,p), z', p\right) - \varphi\left(x(z,p), z, p\right)\right] + \left[\varphi\left(x(z,p), z, p\right) - \varphi\left(x(z',p), z, p\right)\right]$$
$$+ \left[\varphi\left(x(z',p), z, p\right) - \varphi\left(x(z',p), z', p\right)\right]$$
$$= \ \frac{\sigma}{2}[\|x(z,p) - z'\|^2 - \|x(z,p) - z\|^2] + \left[\varphi\left(x(z,p), z, p\right) - \varphi\left(x(z',p), z, p\right)\right]$$
$$+ \frac{\sigma}{2}[\|x(z',p) - z\|^2 - \|x(z',p) - z'\|^2]$$
$$= \ \left[\varphi\left(x(z,p), z, p\right) - \varphi\left(x(z',p), z, p\right)\right] + \sigma\langle z' - z, x(z',p) - x(z,p)\rangle$$
$$\leq \ -\frac{\sigma - (L_f + \rho_g)}{2}\|x(z,p) - x(z',p)\|^2 + \sigma\langle z' - z, x(z',p) - x(z,p)\rangle$$

Again using the strongly convex of $\varphi(x,z,p)$ in $x$, we have that

$$\varphi\left(x(z,p), z', p\right) - \varphi\left(x(z',p), z', p\right) \geq \frac{\sigma - (L_f + \rho_g)}{2}\|x(z,p) - x(z',p)\|^2.$$

Therefore, $\|x(z,p) - x(z',p)\| \leq \frac{\sigma}{\sigma - (L_f + \rho_g)}\|z - z'\|$. $\qquad\square$

**Proposition A.2.** *Suppose $\sigma > L_f + \rho_g$. Let $\{w^k\}$ be generated by N-RPDC. For given $w^k$, the primal output $(x^{k+1}, z^{k+1})$ are random variables, and the following assertion hold:*

$$\|x^k - x(z^k, T_p(w^k))\| \leq \left(\frac{L_f + \sigma + \gamma\|A\|^2 + \frac{L_K}{\alpha_x}}{\sigma - L_f - \rho_g} + 1\right)\|x^k - T_x(w^k)\|. \qquad (18)$$

*Proof.* Since $x(z^k, T_p(w^k)) = \arg\min_{x \in \mathbb{X}} \mathcal{L}_\gamma^+(x, z^k, T_p(w^k))$ and $\mathcal{L}_\gamma^+(x,z,p)$ is $\sigma - L_f - \rho_g$-strongly convex in $x$, then $\forall \zeta \in \partial g(T_x(w^k)) + \mathcal{N}_{\mathbb{X}}(T_x(w^k))$ we have that

$$(\sigma - L_f - \rho_g)\|T_x(w^k) - x(z^k, T_p(w^k))\|^2$$
$$\leq \ \langle \nabla f(T_x(w^k)) + \sigma(T_x(w^k) - z^k) + \zeta, T_x(w^k) - x(z^k, T_p(w^k))\rangle$$
$$+ \langle A^\top[T_p(w^k) + \gamma(AT_x(w^k) - b)], T_x(w^k) - x(z^k, T_p(w^k))\rangle.$$

It follows that

$$(\sigma - L_f - \rho_g)\|T_x(w^k) - x(z^k, T_p(w^k))\|$$
$$\leq \ \|\nabla f(T_x(w^k)) + \sigma(T_x(w^k) - z^k) + \zeta + A^\top[T_p(w^k) + \gamma(AT_x(w^k) - b)]\|.$$

Again using the optimal condition of

$$T_x(w^k) = \arg\min_{x \in \mathbb{X}} \langle \nabla f(x^k) + \sigma(x^k - z^k), x \rangle + g(x) + \langle T_p(w^k) + \gamma(Ax^k - b), Ax \rangle + \frac{D(x,x^k)}{\alpha_x},$$

we have that

$$\begin{aligned}
0 \quad \in \quad & \nabla f(x^k) + \partial g(T_x(w^k)) + \sigma(x^k - z^k) + \mathcal{N}_{\mathbb{X}}(T_x(w^k)) + A^\top[T_p(w^k) + \gamma(Ax^k - b)] \\
& + \frac{1}{\alpha_x}[\nabla K(T_x(w^k)) - \nabla K(x^k)].
\end{aligned}$$

Therefore

$$\begin{aligned}
\xi \quad \in \quad & \nabla f(T_x(w^k)) + \sigma(T_x(w^k) - z^k) + \partial g(T_x(w^k)) + \mathcal{N}_{\mathbb{X}}(T_x(w^k)) \\
& + A^\top[T_p(w^k) + \gamma(AT_x(w^k) - b)],
\end{aligned}$$

with

$\xi = \nabla f(T_x(w^k)) - \nabla f(x^k) + \sigma(T_x(w^k) - x^k) + \gamma A^\top A(T_x(w^k) - x^k) - \frac{1}{\alpha_x}[\nabla K(T_x(w^k)) - \nabla K(x^k)].$

By Assumptions on (P), we obtain that

$$\|\xi\| \le (L_f + \sigma + \gamma\|A\|^2 + \frac{L_K}{\alpha_x})\|x^k - T_x(w^k)\|,$$

and

$$\|T_x(w^k) - x(z^k, T_p(w^k))\| \le \frac{L_f + \sigma + \gamma\|A\|^2 + \frac{L_K}{\alpha_x}}{\sigma - L_f - \rho_g}\|x^k - T_x(w^k)\|.$$

Then by the triangle inequality, above inequality yields the desired result. $\qquad\square$

With these two propositions in hands, we can provide the technical proof of Lemma 3.1.

**Proof of Lemma 3.1:**

**Step 1: Estimate the term $\mathcal{L}_\gamma^+(w^k) - \mathbb{E}_{i(k)}\mathcal{L}_\gamma^+(w^{k+1})$.** By the dual update of N-RPDC, we have that

$$\mathcal{L}_\gamma^+(x^k, z^k, p^k) - \mathcal{L}_\gamma^+(x^k, z^k, p^{k+1}) = \langle p^k - p^{k+1}, Ax^k - b \rangle = -\frac{1}{\eta}\|p^k - p^{k+1}\|^2. \tag{19}$$

Since $x^{k+1}$ is the solution of subproblem of $x$ in N-RPDC scheme, we have that

$$\langle \nabla_{i(k)} f(x^k) + \sigma(x^k - z^k)_{i(k)}, (x - x^{k+1})_{i(k)} \rangle + g_{i(k)}(x_{i(k)}) - g_{i(k)}(x_{i(k)}^{k+1})$$

$$+ \langle p^{k+1} + \gamma(Ax^k - b), A_{i(k)}(x - x^{k+1})_{i(k)} \rangle + \frac{1}{\alpha_x}D(x, x^k) - \frac{1}{\alpha_x}D(x^{k+1}, x^k) \ge 0 \tag{20}$$

Take $x = x^k$ in (20), by the fact that $g_{i(k)}(x_{i(k)}^k) - g_{i(k)}(x_{i(k)}^{k+1}) = g(x^k) - g(x^{k+1})$

and

$$\begin{aligned}
& \langle \nabla_{i(k)} f(x^k) + \sigma(x^k - z^k)_{i(k)} + (A_{i(k)})^\top[p^{k+1} + \gamma(Ax^k - b)], (x^k - x^{k+1})_{i(k)} \rangle \\
= \quad & \langle \nabla f(x^k) + \sigma(x^k - z^k) + A^\top[p^{k+1} + \gamma(Ax^k - b)], x^k - x^{k+1} \rangle,
\end{aligned}$$

we have that

$$\langle \nabla f(x^k) + \sigma(x^k - z^k), x^k - x^{k+1} \rangle + g(x^k) - g(x^{k+1}) + \langle p^{k+1} + \gamma(Ax^k - b), A(x^k - x^{k+1}) \rangle$$

$$-\frac{1}{\alpha_x}D(x^{k+1}, x^k) \ge 0$$

It yields that

$$\begin{aligned}
& F^+(x^k, z^k) - F^+(x^{k+1}, z^k) + \langle p^{k+1} + \gamma(Ax^k - b), A(x^k - x^{k+1}) \rangle \\
\ge \quad & \frac{1}{\alpha_x}D(x^{k+1}, x^k) - [f(x^{k+1}) - f(x^k) - \langle \nabla f(x^k), x^{k+1} - x^k \rangle] - \frac{\sigma}{2}\|x^k - x^{k+1}\|^2 \\
\ge \quad & \frac{\beta_K - \alpha_x(L_f + \sigma)}{2\alpha_x}\|x^k - x^{k+1}\|^2
\end{aligned}$$

Since $\langle\gamma(Ax^k - b), A(x^k - x^{k+1})\rangle = \frac{\gamma}{2}\left(\|Ax^k - b\|^2 - \|Ax^{k+1} - b\|^2 + \|A(x^k - x^{k+1})\|^2\right)$, therefore

$$\mathcal{L}_\gamma^+(x^k, z^k, p^{k+1}) - \mathcal{L}_\gamma^+(x^{k+1}, z^k, p^{k+1}) \geq \frac{\beta_K - \alpha_x(L_f + \sigma + \gamma\|A\|^2)}{2\alpha_x}\|x^k - x^{k+1}\|^2. \quad (21)$$

Additionally, since $z^{k+1}$ is the solution of subproblem of $z$ in N-RPDC, we have that

$$\langle\sigma(z^k - x^k)_{i(k)}, (z - z^{k+1})_{i(k)}\rangle + \frac{1}{\alpha_z}\langle z^{k+1} - z^k, z - z^{k+1}\rangle \geq 0. \quad (22)$$

Take $z = z^k$ in (22), by the fact that $\langle\sigma(z^k - x^k)_{i(k)}, (z^k - z^{k+1})_{i(k)}\rangle = \langle\sigma(z^k - x^k), z^k - z^{k+1}\rangle$, we have that

$$\langle\sigma(z^k - x^k), z^k - z^{k+1}\rangle \geq \frac{1}{\alpha_z}\|z^k - z^{k+1}\|^2.$$

Since

$$\begin{aligned}
\langle\sigma(z^k - x^k), z^k - z^{k+1}\rangle &= \langle\sigma(z^k - x^{k+1}), z^k - z^{k+1}\rangle + \langle\sigma(x^{k+1} - x^k), z^k - z^{k+1}\rangle \\
&= \frac{\sigma}{2}\left(\|x^{k+1} - z^k\|^2 - \|x^{k+1} - z^{k+1}\|^2 + \|z^k - z^{k+1}\|^2\right) \\
&\quad + \langle\sigma(x^{k+1} - x^k), z^k - z^{k+1}\rangle \\
&\leq \frac{\sigma}{2}\left(\|x^{k+1} - z^k\|^2 - \|x^{k+1} - z^{k+1}\|^2 + \|z^k - z^{k+1}\|^2\right) \\
&\quad + \frac{\sigma}{2}\left(\|x^k - x^{k+1}\|^2 + \|z^k - z^{k+1}\|^2\right) \\
&= \frac{\sigma}{2}\left(\|x^{k+1} - z^k\|^2 - \|x^{k+1} - z^{k+1}\|^2\right) + \sigma\|z^k - z^{k+1}\|^2 \\
&\quad + \frac{\sigma}{2}\|x^k - x^{k+1}\|^2,
\end{aligned}$$

above inequality yields that

$$\begin{aligned}
\mathcal{L}_\gamma^+(x^{k+1}, z^k, p^{k+1}) - \mathcal{L}_\gamma^+(x^{k+1}, z^{k+1}, p^{k+1}) &= \frac{\sigma}{2}\left(\|x^{k+1} - z^k\|^2 - \|x^{k+1} - z^{k+1}\|^2\right) \\
&\geq (\frac{1}{\alpha_z} - \sigma)\|z^k - z^{k+1}\|^2 - \frac{\sigma}{2}\|x^k - x^{k+1}\|^2.
\end{aligned} \quad (23)$$

By the combination of (19), (21) and (23), we obtain that

$$\begin{aligned}
&\mathcal{L}_\gamma^+(x^k, z^k, p^k) - \mathcal{L}_\gamma^+(x^{k+1}, z^{k+1}, p^{k+1}) \\
&\geq \frac{\frac{\beta_K}{\alpha_x} - (L_f + 2\sigma + \gamma\|A\|^2)}{2}\|x^k - x^{k+1}\|^2 + (\frac{1}{\alpha_z} - \sigma)\|z^k - z^{k+1}\|^2 - \frac{1}{\eta}\|p^k - p^{k+1}\|^2.
\end{aligned}$$

Take expectation with respect to $i(k)$ on both side of above inequality, we have that

$$\begin{aligned}
&\mathcal{L}_\gamma^+(x^k, z^k, p^k) - \mathbb{E}_{i(k)}\mathcal{L}_\gamma^+(x^{k+1}, z^{k+1}, p^{k+1}) \\
&\geq \frac{\frac{\beta_K}{\alpha_x} - (L_f + 2\sigma + \gamma\|A\|^2)}{2}\mathbb{E}_{i(k)}\|x^k - x^{k+1}\|^2 + (\frac{1}{\alpha_z} - \sigma)\mathbb{E}_{i(k)}\|z^k - z^{k+1}\|^2 \\
&\quad - \frac{1}{\eta}\|p^k - p^{k+1}\|^2.
\end{aligned}$$

By the fact that $\mathbb{E}_{i(k)}\|x^k - x^{k+1}\|^2 = \frac{1}{N}\|x^k - T_x(w^k)\|^2$, $\mathbb{E}_{i(k)}\|z^k - z^{k+1}\|^2 = \frac{1}{N}\|z^k - T_z(w^k)\|^2$ and $p^{k+1} = T_p(w^k)$, above inequality follows that

$$\begin{aligned}
&\mathcal{L}_\gamma^+(x^k, z^k, p^k) - \mathbb{E}_{i(k)}\mathcal{L}_\gamma^+(x^{k+1}, z^{k+1}, p^{k+1}) \\
&\geq \frac{\frac{\beta_K}{\alpha_x} - (L_f + 2\sigma + \gamma\|A\|^2)}{2N}\|x^k - T_x(w^k)\|^2 + \frac{1}{N}(\frac{1}{\alpha_z} - \sigma)\|z^k - T_z(w^k)\|^2 \\
&\quad - \frac{1}{\eta}\|p^k - T_p(w^k)\|^2.
\end{aligned}$$

**Step 2: Estimate the term $\nu(z^k) - \nu(z^{k+1})$.** By the Danskin's theorem, we have that $\nabla\nu(z) = \sigma(z - x(z))$. By statement (i) of Proposition A.1, we have that

$$\|\nabla\nu(z^k) - \nabla\nu(z^{k+1})\| \leq \left(\frac{\sigma^2}{\sigma - L_f - \rho_g} + \sigma\right)\|z^k - z^{k+1}\|.$$

The gradient Lipschitz property of $\nu$ follows that

$$\nu(z^k) - \nu(z^{k+1}) \geq \sigma\langle z^k - x(z^k), z^k - z^{k+1}\rangle - \left(\frac{\sigma^2}{2(\sigma - L_f - \rho_g)} + \frac{\sigma}{2}\right)\|z^k - z^{k+1}\|^2. \quad (24)$$

**Step 3: Estimate the term $\psi_\gamma(z^{k+1}, p^{k+1}) - \psi_\gamma(z^k, p^k)$.** Since $x(z^k, p^k) = \arg\min_{x\in\mathbb{X}}\mathcal{L}_\gamma^+(x, z^k, p^k)$, we have

$$
\begin{aligned}
&\psi_\gamma(z^{k+1}, p^{k+1}) - \psi_\gamma(z^k, p^{k+1})\\
=\ &\mathcal{L}_\gamma^+(x(z^{k+1}, p^{k+1}), z^{k+1}, p^{k+1}) - \mathcal{L}_\gamma^+(x(z^k, p^{k+1}), z^k, p^{k+1})\\
\geq\ &\mathcal{L}_\gamma^+(x(z^{k+1}, p^{k+1}), z^{k+1}, p^{k+1}) - \mathcal{L}_\gamma^+(x(z^{k+1}, p^{k+1}), z^k, p^{k+1})\\
=\ &\frac{\sigma}{2}\left(\|x(z^{k+1}, p^{k+1}) - z^{k+1}\|^2 - \|x(z^{k+1}, p^{k+1}) - z^k\|^2\right)\\
\geq\ &\sigma\langle z^k - x(z^{k+1}, p^{k+1}), z^{k+1} - z^k\rangle.
\end{aligned}
\quad (25)
$$

By the concavity of $\psi_\gamma(z, \cdot)$ and $\nabla\psi_\gamma(z, p) = Ax(z, p) - b$, we have

$$
\begin{aligned}
\psi_\gamma(z^k, p^{k+1}) - \psi_\gamma(z^k, p^k) &\geq \langle p^{k+1} - p^k, Ax(z^k, p^{k+1}) - b\rangle\\
&= \frac{1}{2\eta}\|p^k - p^{k+1}\|^2 + \frac{\eta}{2}\|Ax(z^k, p^{k+1}) - b\|^2\\
&\quad - \frac{1}{2\eta}\|p^{k+1} - p^k - \eta[Ax(z^k, p^{k+1}) - b]\|^2.
\end{aligned}
$$

By the dual update of N-RPDC, the above inequality yields that

$$
\begin{aligned}
&\psi_\gamma(z^k, p^{k+1}) - \psi_\gamma(z^k, p^k)\\
\geq\ &\frac{1}{2\eta}\|p^k - p^{k+1}\|^2 + \frac{\eta}{2}\|Ax(z^k, p^{k+1}) - b\|^2 - \frac{1}{2\eta}\|\eta A[x^k - x(z^k, p^{k+1})]\|^2\\
\geq\ &\frac{1}{2\eta}\|p^k - p^{k+1}\|^2 + \frac{\eta}{2}\|Ax(z^k, p^{k+1}) - b\|^2 - \frac{\eta}{2}\|A\|^2 \cdot \|x^k - x(z^k, p^{k+1})\|^2. \quad (26)
\end{aligned}
$$

By Proposition A.2, (26) guarantees that

$$
\begin{aligned}
\psi_\gamma(z^k, p^{k+1}) - \psi_\gamma(z^k, p^k) \geq\ &\frac{1}{2\eta}\|p^k - p^{k+1}\|^2 + \frac{\eta}{2}\|Ax(z^k, p^{k+1}) - b\|^2\\
&- \frac{\eta}{2}\|A\|^2\left(\frac{L_f + \sigma + \gamma\|A\|^2 + \frac{L_K}{\alpha_x}}{\sigma - L_f - \rho_g} + 1\right)^2\|x^k - T_x(w^k)\|^2.
\end{aligned}
\quad (27)
$$

By the combination of (25) and (27), we obtain that

$$
\begin{aligned}
&\psi_\gamma(z^{k+1}, p^{k+1}) - \psi_\gamma(z^k, p^k)\\
\geq\ &\sigma\langle z^k - x(z^{k+1}, p^{k+1}), z^{k+1} - z^k\rangle + \frac{1}{2\eta}\|p^k - p^{k+1}\|^2\\
&+ \frac{\eta}{2}\|Ax(z^k, p^{k+1}) - b\|^2 - \frac{\eta}{2}\|A\|^2\left(\frac{L_f + \sigma + \gamma\|A\|^2 + \frac{L_K}{\alpha_x}}{\sigma - L_f - \rho_g} + 1\right)^2\|x^k - T_x(w^k)\|^2.
\end{aligned}
\quad (28)
$$

**Step 4: Estimate the term** $[\nu(z^k) - \psi_\gamma(z^k, p^k)] - \mathbb{E}_{i(k)}[\nu(z^{k+1}) - \psi_\gamma(z^{k+1}, p^{k+1})]$. Summing statement (24) and (28), we obtain the variation of dual gap:

$$[\nu(z^k) - \psi_\gamma(z^k, p^k)] - [\nu(z^{k+1}) - \psi_\gamma(z^{k+1}, p^{k+1})]$$

$$\geq \sigma\langle x(z^{k+1}, p^{k+1}) - x(z^k), z^k - z^{k+1}\rangle - \left(\frac{\sigma^2}{2(\sigma - L_f - \rho_g)} + \frac{\sigma}{2}\right)\|z^k - z^{k+1}\|^2$$

$$+ \frac{1}{2\eta}\|p^k - p^{k+1}\|^2 + \frac{\eta}{2}\|Ax(z^k, p^{k+1}) - b\|^2$$

$$- \frac{\eta}{2}\|A\|^2\left(\frac{L_f + \sigma + \gamma\|A\|^2 + \frac{L_K}{\alpha_x}}{\sigma - L_f - \rho_g} + 1\right)^2\|x^k - T_x(w^k)\|^2. \tag{29}$$

By Proposition A.1, we have that

$$\sigma\langle x(z^{k+1}, p^{k+1}) - x(z^k), z^k - z^{k+1}\rangle$$

$$= \sigma[\langle x(z^k, p^{k+1}) - x(z^k), z^k - z^{k+1}\rangle + \langle x(z^{k+1}, p^{k+1}) - x(z^k, p^{k+1}), z^k - z^{k+1}\rangle]$$

$$\geq -\frac{\sigma}{2\lambda}\|z^k - z^{k+1}\|^2 - \frac{\sigma\lambda}{2}\|x(z^k, p^{k+1}) - x(z^k)\|^2$$

$$- \sigma\|z^k - z^{k+1}\| \cdot \|x(z^k, p^{k+1}) - x(z^{k+1}, p^{k+1})\|$$

$$\geq -\left(\frac{\sigma}{2\lambda} + \frac{\sigma^2}{\sigma - L_f - \rho_g}\right)\|z^k - z^{k+1}\|^2 - \frac{\sigma\lambda}{2}\|x(z^k, p^{k+1}) - x(z^k)\|^2, \tag{30}$$

with $\lambda > 0$ is any positive number. By the combination of (29) and (30), we have that

$$[\nu(z^k) - \psi_\gamma(z^k, p^k)] - [\nu(z^{k+1}) - \psi_\gamma(z^{k+1}, p^{k+1})]$$

$$\geq -\left(\frac{\sigma}{2\lambda} + \frac{3\sigma^2}{2(\sigma - L_f - \rho_g)} + \frac{\sigma}{2}\right)\|z^k - z^{k+1}\|^2 - \frac{\sigma\lambda}{2}\|x(z^k, p^{k+1}) - x(z^k)\|^2$$

$$+ \frac{1}{2\eta}\|p^k - p^{k+1}\|^2 + \frac{\eta}{2}\|Ax(z^k, p^{k+1}) - b\|^2$$

$$- \frac{\eta}{2}\|A\|^2\left(\frac{L_f + \sigma + \gamma\|A\|^2 + \frac{L_K}{\alpha_x}}{\sigma - L_f - \rho_g} + 1\right)^2\|x^k - T_x(w^k)\|^2. \tag{31}$$

Take expectation with respect to $i(k)$ on both side of above inequality, we have that

$$[\nu(z^k) - \psi_\gamma(z^k, p^k)] - \mathbb{E}_{i(k)}[\nu(z^{k+1}) - \psi_\gamma(z^{k+1}, p^{k+1})]$$

$$\geq -\left(\frac{\sigma}{2\lambda} + \frac{3\sigma^2}{2(\sigma - L_f - \rho_g)} + \frac{\sigma}{2}\right)\mathbb{E}_{i(k)}\|z^k - z^{k+1}\|^2 - \frac{\sigma\lambda}{2}\|x(z^k, p^{k+1}) - x(z^k)\|^2$$

$$+ \frac{1}{2\eta}\|p^k - p^{k+1}\|^2 + \frac{\eta}{2}\|Ax(z^k, p^{k+1}) - b\|^2$$

$$- \frac{\eta}{2}\|A\|^2\left(\frac{L_f + \sigma + \gamma\|A\|^2 + \frac{L_K}{\alpha_x}}{\sigma - L_f - \rho_g} + 1\right)^2\|x^k - T_x(w^k)\|^2.$$

By the fact that $\mathbb{E}_{i(k)}\|z^k - z^{k+1}\|^2 = \frac{1}{N}\|z^k - T_z(w^k)\|^2$ and $p^{k+1} = T_p(w^k)$, above inequality follows that

$$[\nu(z^k) - \psi_\gamma(z^k, p^k)] - \mathbb{E}_{i(k)}[\nu(z^{k+1}) - \psi_\gamma(z^{k+1}, p^{k+1})]$$

$$\geq -\frac{1}{N}\left(\frac{\sigma}{2\lambda} + \frac{3\sigma^2}{2(\sigma - L_f - \rho_g)} + \frac{\sigma}{2}\right)\|z^k - T_z(w^k)\|^2 - \frac{\sigma\lambda}{2}\|x(z^k, T_p(w^k)) - x(z^k)\|^2$$

$$+ \frac{1}{2\eta}\|p^k - T_p(w^k)\|^2 + \frac{\eta}{2}\|Ax(z^k, T_p(w^k)) - b\|^2$$

$$- \frac{\eta}{2}\|A\|^2\left(\frac{L_f + \sigma + \gamma\|A\|^2 + \frac{L_K}{\alpha_x}}{\sigma - L_f - \rho_g} + 1\right)^2\|x^k - T_x(w^k)\|^2. \tag{32}$$

**Step 5: Estimate the term $\Lambda(w^k) - \mathbb{E}_{i(k)}\Lambda(w^{k+1})$.** By the combination of Step 1 and Step 4, we obtain that

$$
\Lambda(w^k) - \mathbb{E}_{i(k)}\Lambda(w^{k+1})
$$

$$
\geq \left[\frac{\frac{\beta_K}{\alpha_x} - (L_f + 2\sigma + \gamma\|A\|^2)}{2N} - \eta\|A\|^2\left(\frac{L_f + \sigma + \gamma\|A\|^2 + \frac{L_K}{\alpha_x}}{\sigma - L_f - \rho_g} + 1\right)^2\right]\|x^k - T_x(w^k)\|^2
$$

$$
+\frac{1}{N}\left[\frac{1}{\alpha_z} - 2\sigma - \frac{\sigma}{\lambda} - \frac{3\sigma^2}{\sigma - L_f - \rho_g}\right]\|z^k - T_z(w^k)\|^2
$$

$$
+\eta\|Ax(z^k, T_p(w^k)) - b\|^2 - \sigma\lambda\|x(z^k, T_p(w^k)) - x(z^k)\|^2. \tag{33}
$$

Since $0 < \alpha_x \leq \beta_K/[L_f + 2\sigma + \gamma\|A\|^2 + 5]$, then $\frac{\beta_K}{\alpha_x} - (L_f + 2\sigma + \gamma\|A\|^2) \geq 5$. Moreover, $0 < \eta \leq 1/\left[2N\|A\|^2\left(\frac{L_f + \sigma + \gamma\|A\|^2 + \frac{L_K}{\alpha_x}}{\sigma - L_f - \rho_g} + 1\right)^2\right]$ implies

$$
\Lambda(w^k) - \mathbb{E}_{i(k)}\Lambda(w^{k+1})
$$

$$
\geq \frac{2}{N}\|x^k - T_x(w^k)\|^2 + \frac{1}{N}\left[\frac{1}{\alpha_z} - 2\sigma - \frac{\sigma}{\lambda} - \frac{3\sigma^2}{\sigma - L_f - \rho_g}\right]\|z^k - T_z(w^k)\|^2
$$

$$
+\eta\|Ax(z^k, T_p(w^k)) - b\|^2 - \sigma\lambda\|x(z^k, T_p(w^k)) - x(z^k)\|^2. \tag{34}
$$

### A.6   PROOF OF LEMMA 3.2

*Proof.* The result is directly by the fact that $dist\left(0, \partial\mathcal{L}_\gamma^z\left(x(z,p), p\right)\right) = \|Ax(z,p) - b\|$.   □

### A.7   PROOF OF LEMMA 3.3

*Proof.* The claim is provided by the results of Proposition 1 and Corollary of (Robinson, 1981).   □

### A.8   PROOF OF LEMMA 4.1

*Proof.* Taking $\lambda = \min\{\frac{\delta\eta}{2\sigma M^2+1}, \frac{\eta}{\sigma\kappa^2+1}\}$ and $0 < \alpha_z < 1/\left[2\sigma + \frac{\sigma}{\lambda} + \frac{3\sigma^2}{\sigma - L_f - \rho_g} + 1\right]$ in Lemma 3.1, we have that

$$
\Lambda(w^k) - \mathbb{E}_{i(k)}\Lambda(w^{k+1})
$$

$$
\geq \frac{2}{N}\|x^k - T_x(w^k)\|^2 + \frac{1}{N}\|z^k - T_z(w^k)\|^2 + \eta\|Ax(z^k, T_p(w^k)) - b\|^2
$$

$$
-\sigma\min\{\frac{\delta\eta}{2\sigma M^2 + 1}, \frac{\eta}{\sigma\kappa^2 + 1}\}\|x(z^k, T_p(w^k)) - x(z^k)\|^2. \tag{35}
$$

For the last two terms of above inequality, we have two cases to discuss.

- Case 1: $\|Ax(z^k, T_p(w^k)) - b\| \leq \frac{2\sigma\lambda M^2}{\eta} \leq \delta$.

  Since $\partial\mathcal{L}_\gamma^z(x,p)$ is locally metric subregular in $(x,p)$ uniformly in $z$ over $\mathbb{R}^d$ at point $0$ with parameters $\kappa, \delta > 0$, (35) yields that

$$
\Lambda(w^k) - \mathbb{E}_{i(k)}\Lambda(w^{k+1}) \geq \frac{2}{N}\|x^k - T_x(w^k)\|^2 + \frac{1}{N}\|z^k - T_z(w^k)\|^2
$$

$$
+\frac{\eta}{\sigma\kappa^2 + 1}\|Ax(z^k, T_p(w^k)) - b\|^2. \tag{36}
$$

- Case 2: $\|Ax(z^k, T_p(w^k)) - b\| > \frac{2\sigma\lambda M^2}{\eta}$.

Since $M = \max\limits_{x,x' \in \mathbb{X}} \|x - x'\|$, we have that

$$
\begin{aligned}
&\eta\|Ax(z^k, T_p(w^k)) - b\|^2 - \sigma\lambda\|x(z^k, T_p(w^k)) - x^*(z^k)\|^2 \\
\geq\ & \frac{\eta}{2}\|Ax(z^k, T_p(w^k)) - b\|^2 + \frac{\eta}{2} \cdot \frac{2\sigma\lambda M^2}{\eta} - \sigma\lambda M^2 \\
=\ & \frac{\eta}{2}\|Ax(z^k, T_p(w^k)) - b\|^2.
\end{aligned}
\tag{37}
$$

For both case, (35) yields that

$$
\begin{aligned}
&\Lambda(w^k) - \mathbb{E}_{i(k)}\Lambda(w^{k+1}) \\
\geq\ & \frac{2}{N}\|x^k - T_x(w^k)\|^2 + \frac{1}{N}\|z^k - T_z(w^k)\|^2 + \frac{\eta}{\sigma\kappa^2 + 2}\|Ax(z^k, T_p(w^k)) - b\|^2.
\end{aligned}
\tag{38}
$$

By Proposition A.2, above inequality yields that

$$
\begin{aligned}
&\Lambda(w^k) - \mathbb{E}_{i(k)}\Lambda(w^{k+1}) \\
\geq\ & \frac{1}{N}\|x^k - T_x(w^k)\|^2 + \frac{1}{N\left(\frac{L_f + \sigma + \gamma\|A\|^2 + \frac{L_K}{\alpha_x}}{\sigma - L_f - \rho_g} + 1\right)^2}\|x^k - x(z^k, T_p(w^k))\|^2 \\
&+ \frac{1}{N}\|z^k - T_z(w^k)\|^2 + \frac{\eta}{\sigma\kappa^2 + 2}\|Ax(z^k, T_p(w^k)) - b\|^2 \\
\geq\ & \frac{1}{N}\|x^k - T_x(w^k)\|^2 + \frac{1}{N\|A\|^2\left(\frac{L_f + \sigma + \gamma\|A\|^2 + \frac{L_K}{\alpha_x}}{\sigma - L_f - \rho_g} + 1\right)^2}\|A(x^k - x(z^k, T_p(w^k)))\|^2 \\
&+ \frac{1}{N}\|z^k - T_z(w^k)\|^2 + \frac{\eta}{\sigma\kappa^2 + 2}\|Ax(z^k, T_p(w^k)) - b\|^2 \\
\geq\ & \frac{1}{N}\|x^k - T_x(w^k)\|^2 + \frac{1}{N}\|z^k - T_z(w^k)\|^2 \\
&+ \frac{\eta}{2(\sigma\kappa^2 + 2)}\left(2\|A(x^k - x(z^k, T_p(w^k)))\|^2 + 2\|Ax(z^k, T_p(w^k)) - b\|^2\right) \\
\geq\ & \frac{1}{N}\|x^k - T_x(w^k)\|^2 + \frac{1}{N}\|z^k - T_z(w^k)\|^2 + \frac{\eta}{2(\sigma\kappa^2 + 2)}\|Ax^k - b\|^2 \\
=\ & \frac{1}{N}\|x^k - T_x(w^k)\|^2 + \frac{1}{N}\|z^k - T_z(w^k)\|^2 + \frac{1}{2\eta(\sigma\kappa^2 + 2)}\|p^k - T_p(w^k)\|^2 \\
\geq\ & \frac{c_1}{N}\|w^k - T(w^k)\|^2,
\end{aligned}
\tag{39}
$$

with $c_1 = \min\left\{1, \frac{N}{2\eta(\sigma\kappa^2 + 2) + 1}\right\}$.  $\square$

## A.9  PROOF OF THEOREM 4.1

In order to provide the Theorem 4.1, we need the famous Robbins-Siegmund theorem.

**Theorem A.1. (Robbins-Siegmund Theorem, Robbins & Siegmund (1971), Theorem 1.3.12 in Duflo (2013), Theorem 2.27 in Carpentier et al. (2015))** *Let $\{\Lambda^k\}_{k \in \mathbb{N}}$, $\{\mu^k\}_{k \in \mathbb{N}}$, $\{\nu^k\}_{k \in \mathbb{N}}$ and $\{\eta^k\}_{k \in \mathbb{N}}$ be four positive sequences of real-valued random variables adapted to the filtration $\{\xi_k\}_{k \in \mathbb{N}}$. Assume that*

$$
\mathbb{E}_{\xi_k}\Lambda^{k+1} \leq (1 + \mu^k)\Lambda^k + \nu^k - \eta^k, \quad \forall k \in \mathbb{N},
$$

*and that*

$$
\sum_{k \in \mathbb{N}} \mu^k < +\infty \quad and \quad \sum_{k \in \mathbb{N}} \nu^k < +\infty, \quad a.s..
$$

*Then, the sequence $\{\Lambda^k\}_{k\in\mathbb{N}}$ almost surely converges to a finite[3] random variable $\Lambda^\infty$, and $\sum\limits_{k\in\mathbb{N}} \eta^k < +\infty$, a.s..*

Then we can derive the technical proof of this theorem.

**Proof of Theorem 4.1:**

(a) By Lemma 4.1 and the fact that $\Lambda(w) \geq F^*$, we have that

$$\mathbb{E}_{i(k)}[\Lambda(w^{k+1}) - F^*] \leq [\Lambda(w^k) - F^*] - \frac{c_1}{N}\|w^k - T(w^k)\|^2.$$

By the Robbins-Siegmund Theorem (Theorem A.1), we obtain that, $\lim\limits_{k\to+\infty} \Lambda(w^k)$ is almost surely exists and $\sum\limits_{k=0}^{+\infty} \|w^k - T(w^k)\|^2 < +\infty$ a.s.. Therefore, we have $\lim_{k\to+\infty} \Phi(w^k) = 0$;

(b) By the compact of $\mathbb{X}$, we have the sequences $\{x^k\}$ and $\{x(z^k)\}$ are bounded.

By statement (a), we have $\lim\limits_{k\to+\infty} \Lambda(w^k)$ is almost surely exists. It follows that the sequence $\{\Lambda(w^k)\}$ is almost surely bounded. By the definition of $\Lambda(w^k)$, we also have that $\Lambda(w^k) \geq \nu(z^k) \geq F(x(z^k)) \geq F^*$. Therefore the sequence $\{\nu(z^k)\}$ is almost surely bounded.

Now we show the almost surely boundness of the sequence $\{z^k\}$ by contradiction. Suppose $\mathbb{P}\{\|z^k\| \to +\infty\} > 0$. Since the sequence $\{x(z^k)\}$ is bounded, there exists a subsequence $\{z^{k'}\}$ such that $x(z^{k'}) \to \bar{x}$. Therefore $\mathbb{P}\{\nu(z^{k'}) = F(x(z^{k'})) + \frac{\sigma}{2}\|(x(z^{k'})) - z^{k'}\|^2 \to +\infty\} > 0$, which follows a contradiction with that $\{\nu(z^{k'})\}$ is almost surely bounded. Therefore, we obtain the almost surely boundness of the sequence $\{z^k\}$.

Next we propose the boundness of $\{p^k\}$. By the optimal condition of problem of $T_x(w^k) = \arg\min\limits_{x\in\mathbb{X}} \langle \nabla f(x^k) + \sigma(x^k - z^k), x\rangle + g(x) + \langle T_p(w^k) + \gamma(Ax^k - b), Ax\rangle + \frac{D(x,x^k)}{\alpha_x}$, we have

$$0 \in \nabla f(x^k) + \sigma(x^k - z^k) + \partial g(T_x(w^k)) + \mathcal{N}_{\mathbb{X}}(T_x(w^k)) + A^\top[T_p(w^k) + \gamma(Ax^k - b)]$$
$$+ \frac{1}{\alpha_x}[\nabla K(T_x(w^k)) - \nabla K(x^k)].$$

Since $T_x(w^k) = N\mathbb{E}_{i(k)}x^{k+1} - (N-1)x^k$ and the boundness of the sequence $\{x^k\}$, we have the sequence $\{T_x(w^k)\}$ is bounded. By bounded of subgradient of $g$, the boundness of $\{x^k\}, \{T_x(w^k)\}$ and the almost surely boundness of $\{z^k\}$, it follows that $\|A^\top T_p(w^k)\| < +\infty$, $\forall k > 0$, a.s.. By the fact that $p^{k+1} = T_p(w^k)$, we have that

$$\|A^\top p^{k+1}\| < +\infty, \quad \forall k > 0, \quad a.s.$$

By update of $p$ in N-RPDC, we have that

$$p^{k+1} = p^0 + \hat{p}^{k+1} \quad \text{when} \quad \hat{p}^{k+1} = \eta\sum_{j=1}^{k}(Ax^j - b) = \eta A\sum_{j=1}^{k}(x^j - x^*) \in Im(A),$$

with $x^*$ be the optimal solution of (P). Since $p^0 \in \text{Null}(A^\top)$, we have $A^\top p^0 = 0$ and $\|A^\top \hat{p}^{k+1}\| < +\infty$, a.s.. Then we have two cases to discuss.

- Case 1: $rank(A) = n$. Obviously, we have that

$$\|\hat{p}^{k+1}\| \leq \frac{\|A^\top \hat{p}^{k+1}\|}{\sqrt{\lambda_{\min}(AA^\top)}} < +\infty,$$

  with $\lambda_{\min}(AA^\top)$ be the smallest eigenvalue of matrix $AA^\top$.
- Case 2: $rank(A) = r < n$. Without loss of generality, assume the first $r$ rows of $A$ (denoted by $A^r \in \mathbb{R}^{r\times d}$) are linearly independent, we have

$$A = \begin{pmatrix} A^r \\ BA^r \end{pmatrix} = \begin{pmatrix} I_{r\times r} \\ B \end{pmatrix} A^r,$$

---

[3]A random variable $X$ is finite if $\mathbb{P}(\{\omega \in \Omega | X(\omega) = +\infty\}) = 0$.

where $I_{r\times r} \in \mathbb{R}^{r\times r}$ is the indentity matrix and $B \in \mathbb{R}^{(n-r)\times r}$. Let $Q := I_{r\times r} + B^\top B$. It is easy to show that $Q$ is symmetric and positive definite. By the fact that $\hat{p}^{k+1} = \eta A \sum_{j=1}^{k}(x^j - x^*)$, we have that

$$
\begin{aligned}
\|A^\top \hat{p}^{k+1}\|^2 &= \left\| (A^r)^\top \left(I_{r\times r}, B^\top\right) \eta \begin{pmatrix} I_{r\times r} \\ B \end{pmatrix} A^r \sum_{j=1}^{k}(x^j - x^*) \right\|^2 \\
&= \left\| (A^r)^\top Q \eta A^r \sum_{j=1}^{k}(x^j - x^*) \right\|^2 \\
&\geq \lambda_{\min}\left(A^r (A^r)^\top\right) \left\| Q \eta A^r \sum_{j=1}^{k}(x^j - x^*) \right\|^2 \\
&\geq \lambda_{\min}\left(A^r (A^r)^\top\right) \lambda_{\min}(Q^\top Q) \| \eta A^r \sum_{j=1}^{k}(x^j - x^*) \|^2,
\end{aligned}
$$

where $\lambda_{\min}\left(A^r (A^r)^\top\right)$ is the smallest eigenvalue of positive semidefinite matrix $A^r (A^r)^\top$ and $\lambda_{\min}(Q^\top Q)$ is the smallest eigenvalue of positive semidefinite matrix $Q^\top Q$. Using the fact that

$$
\begin{aligned}
\|\hat{p}^{k+1}\|^2 &= \| \eta A \sum_{j=1}^{k}(x^j - x^*) \|^2 \\
&= \| \eta \begin{pmatrix} I_{r\times r} \\ B \end{pmatrix} A^r \sum_{j=1}^{k}(x^j - x^*) \|^2 \\
&\leq \lambda_{\max}(Q) \| \eta A^r \sum_{j=1}^{k}(x^j - x^*) \|^2,
\end{aligned}
$$

where $\lambda_{\max}(Q)$ is the largest eigenvalue of matrix $Q$. Therefore, we have that

$$
\|\hat{p}^{k+1}\|^2 \leq \frac{\lambda_{\max}(Q)}{\lambda_{\min}\left(A^r (A^r)^\top\right) \lambda_{\min}(Q^\top Q)} \|A^\top \hat{p}^{k+1}\|^2.
$$

Therefore, for both cases, there exists $\eth > 0$ such that

$$
\|\hat{p}^{k+1}\| \leq \eth \|A^\top \hat{p}^{k+1}\| < +\infty, a.s.
$$

which yields the almost surely boundness of $\{p^k\}$ by the triangle inequality. Therefore, the sequence $\{w^k\}$ is almost surely bounded.

(c) By $\lim_{k\to+\infty} \|w^k - T(w^k)\| = 0$ $a.s.$ in statment (a), Lemma 2.1, Proposition 3.1, the almost surely boundness of $\{w^k\}$, and the closedness of $\partial \mathcal{L}_\gamma^+(\cdot)$, we obtain the desired result.

## A.10  PROOF OF THEOREM 4.2

*Proof.* Suppose this statement does not hold, that is,

$$
\mathbb{P}\left\{ \liminf_{k\to+\infty} \sqrt{k+1}\Phi(w^k) \geq \delta \right\} > 0,
$$

for some $\eth > 0$. Then for $k_0$ large enough, for all $k \geq k_0$ we have

$$
\mathbb{P}\left\{ \Phi(w^k) \geq \frac{\eth}{\sqrt{k+1}} \right\} > 0.
$$

Therefore

$$
\mathbb{P}\left\{ \sum_{k=k_0}^{+\infty} (\Phi(w^k))^2 \geq \sum_{k=k_0}^{+\infty} (\Phi(w^k))^2 \geq \sum_{k=k_0}^{+\infty} \frac{\eth^2}{k+1} \right\} > 0.
$$

Since the fact that $\sum\limits_{k=k_0}^{+\infty} \frac{\partial^2}{k+1} = +\infty$, we have that above inequality is contradicted with the fact that

$\sum\limits_{k=0}^{+\infty} (\Phi(w^k))^2 < +\infty$ $a.s.$ in the proof of Theorem 4.1. Therefore, we obtain the desired result. $\quad\square$

## A.11 PROOF OF THEOREM 4.3

*Proof.* Recalling the inequality of Lemma 4.1 and the fact that $\Lambda(w^k) \geq \nu(z^k) \geq F^*$, we have

$$\mathbb{E}_{i(k)}[\Lambda(w^{k+1}) - F^*] \leq [\Lambda(w^k) - F^*] - \frac{c_1}{c^2 N}(\Phi(w^k))^2.$$

Taking expectation with respect to $\mathcal{F}_t$, $t > k$ for above inequality, we obtain that

$$\mathbb{E}_{\mathcal{F}_t}[\Lambda(w^{k+1}) - F^*] \leq \mathbb{E}_{\mathcal{F}_t}[\Lambda(w^k) - F^*] - \frac{c_1}{c^2 N}\mathbb{E}_{\mathcal{F}_t}(\Phi(w^k))^2.$$

By the fact $\Lambda(w^k) \geq \nu(z^k) \geq F^*$, it follows

$$\frac{c_1}{c^2 N}\sum_{k=0}^{t} \mathbb{E}_{\mathcal{F}_t}(\Phi(w^k))^2 \leq \Lambda(w^0) - F^*.$$

By the Jensen's inequality and the convexity of function $h(x) = x^2$, $x \in \mathbb{R}$, it follows that

$$\frac{c_1}{c^2 N}\sum_{k=0}^{t} \left(\mathbb{E}_{\mathcal{F}_t}\Phi(w^k)\right)^2 \leq \Lambda(w^0) - F^*.$$

By the fact that $\mathbb{E}_{\mathcal{F}_t}\Phi(w^k) \geq 0$ and $h(x) = x^2$, $x \in \mathbb{R}$ is monotonic increasing in $x$ on $[0, +\infty)$, we have that

$$\frac{c_1}{c^2 N}(t+1)\left(\min_{0 \leq k \leq t} \mathbb{E}_{\mathcal{F}_t}\Phi(w^k)\right)^2 \leq \Lambda(w^0) - F^*.$$

Here comes that

$$\min_{0 \leq k \leq t} \mathbb{E}_{\mathcal{F}_t}\Phi(w^k) \leq \sqrt{\frac{\frac{c^2 N}{c_1}(\Lambda(w^0) - F^*)}{t+1}}.$$

$\quad\square$

