# OpenReview forum: "Randomized Primal-Dual Coordinate Method for Large-scale Linearly Constrained Nonsmooth Nonconvex Optimization"
_ICLR.cc/2022/Conference — ICLR 2022 Submitted_

### Official Review · Reviewer_Wyna · 2021-11-01

**Correctness:** 4
**Technical Novelty And Significance:** 3
**Empirical Novelty And Significance:** 3
**Recommendation:** 6
**Confidence:** 3

**Main Review:**

Strength:
- The paper relaxes the uniform boundedness assumption of the subgradients for general nonconvex functions $g$ to the linear boundedness assumption of the subgradients for a class of weakly convex function $g$.
- To conduct the analysis for the randomized algorithm, a surrogate stationarity measure is proposed based on the distance of the generated sequence of a reference point.

Weakness:
- The paper is merely an extension of Algorithm 2.2 in Zhang & Luo (2020b) to a randomized version. Although some new notations/relaxation of assumptions were proposed, the analysis and techniques of the paper rely on a combination of the techniques used in Zhang & Luo (2020b) and standard techniques of randomized algorithms in the literature.


**Summary Of The Paper:**

The paper proposes a randomized primal-dual method for solving the nonconvex nonsmooth optimization problem with linear constraints. The main analysis relies on the idea of using auxiliary problems and the Lyapunov function proposed in  (Zhang & Luo, 2020b).
The paper shows that any cluster point of the generated sequence is almost surely a stationary point of the nonconvex nonsmooth problem and also provides an almost sure asymptotic convergence rate together with the expected iteration complexity of the proposed method.


**Summary Of The Review:**

The paper is well written and has certain contributions. I would support its publication in ICLR but my support is not strong.

---

> ### Author Response · Authors · 2021-11-13
> **Reply to Reviewer Wyna**
>
> Thanks for your positive evaluation on our work.
>
> Yes, it is true that our proof framework builds upon the pioneering work [1]. However, we have several nontrivial contributions as stated in `Main contributions':
>
> 1) We involve the nonsmooth nonconvex term $g$, which is more general to cover more  problems arising in practice.
>
> 2) We proposed a new algorithm, i.e., N-RPDC, which is more suitably for modern large-scale problems.
>
> 3) Due to the term $g$, we cannot use the dual error bound utilized in [1] for establishing the descent property of N-RPDC. Instead, we identified  the local uniform metric subregularity, which can be used to replace the dual error bound condition for showing descent property of N-RPDC.  We remark that the identification of this new condition may seem simple in retrospect. However, identifying such a condition is actually not that straightforward and rather important in order to show convergence.
>
> 4) As you also mentioned, due to the randomness of N-RPDC, we introduced a new surrogate stationarity measure to characterize the convergence  of N-RPDC.
>
> We hope that these responses are satisfactory to you.
>
> [1] Zhang, J., \& Luo, Z. Q. (2020). A proximal alternating direction method of multiplier for linearly constrained nonconvex minimization. SIAM Journal on Optimization, 30(3), 2272-2302.

---

> ### Author Response · Authors · 2021-11-26
> **To Reviewer Wyna**
>
> Dear Reviewer,
>
> Thanks again for reviewing our manuscript and for your positive evaluation. We are writing to kindly request your feedback on our rebuttals, which we submitted 13 days ago. We believe that we have addressed all your concerns. Specifically, we carefully argued the main differences between our manuscript and the pioneering work you mentioned, which clarifies the novelty of our work. Hence, we hope that our response will be helpful to assist you in re-evaluating our manuscript.
> Please note that the deadline of the author-reviewer discussion stage is approaching. Therefore, please let us know your further views.  If you believe we have not addressed some of your concerns, please let us know why, so that we can either realize there does exist an issue somewhere or we have a chance to clarify further.
>
>
> Best,
>
> Authors.

---

### Official Review · Reviewer_NZQN · 2021-11-02

**Correctness:** 4
**Technical Novelty And Significance:** 2
**Empirical Novelty And Significance:** Not applicable
**Recommendation:** 6
**Confidence:** 3

**Main Review:**

1. For convex case (including nonsmooth and bregman divergence), this paper proposed randomized coordinate primal-dual coordinate method for augmented Lagrangian.

Parallel Direction Method of Multipliers
Huahua Wang, Arindam Banerjee, Zhi-Quan Luo, Neurips, 2014

2.  The z update is cheap and has a closed form, it seems unnecessary to use the gradient and coordinate. I think it will also make the algorithm faster and it may not need a smallest enough step size for the z update as in Lemma 4.1. Further question is if simply using the closed form, could we use ADMM Gauss-Seidel style update, i.e. ||z-x^k || ?  empirically runs faster.

3. In Lemma 3.1, dual step size  \eta requires to be very small. Explanation and tuition could help better understand the algorithm.

4. The paper could present more about the experiments in terms of comparison with existing methods to show the effectiveness of the proposed method, particularly when double the dimension of primal variable.

**Summary Of The Paper:**

This paper studies a particular type of nonconvex nonsmooth problem. The authors propose to use the Bertsekas' convexification procedure on the nonconvex problem, and then use randomized coordinate method to solve the corresponding augmented Lagrangian. The paper also proposes a so-called surrogate stationarity measure to establish the convergence rate for the proposed method.

**Summary Of The Review:**

Overall, the paper looks good and the results are aligned with randomized coordinate methods, i.e. O(1/N).

---

> ### Author Response · Authors · 2021-11-13
> **Reply to Reviewer NZQN**
>
> We appreciate your valuable feedback.
>
> 1. We have added the mentioned reference in the `Related works' in the revision.
>
> 2. Using a gradient-step for the update of $z$ is for the consideration of convergence analysis. It is not direct to see whether we can apply the similar convergence analysis if we update $z$ by using a closed-form solution.      Our Remark 2.1 in the revised version provides that the complexity of each iteration of N-RPDC is $O(\max\\{m,n\\})$ if we update one coordinate of $x$ and $z$ at each iteration. However, updating the whole $z$ at each iteration has complexity of $O(d)$. In this situation, N-RPDC can be meaningless if $d\gg \max\\{m,n\\}$. Therefore, we only  update one block of $z$ at each iteration.
>
> 3. We think small enough dual step size is common in primal-dual methods. This is because primal part seeks to decrease the function value, while the dual part aims to increase the function value. Thus, in order to ensure that the function value is descending in total, we often need the primal is more significant than the dual. We will add this intuition in the final file as it is hard to add it in the current stage (due to the space limitation).
>
> 4. We have added comparison to the existing work [1]; see Fig. 1 in the appendix of the revised version. The difficulty of adding more comparison lies in that there is not other very related existing work (except for [1]) due to our relatively new problem setting; see also our reply to Reviewer Adyx.
>
>
> We hope that these revisions and responses are satisfactory to you.
>
> [1] Zhang, J., \& Luo, Z. Q. (2020). A proximal alternating direction method of multiplier for linearly constrained nonconvex minimization. SIAM Journal on Optimization, 30(3), 2272-2302.

---

> ### Author Response · Authors · 2021-11-26
> **To Reviewer NZQN**
>
> Dear Reviewer,
>
> Thanks again for reviewing our manuscript and for your positive evaluation. We are writing to kindly request your feedback on our rebuttals, which we submitted 13 days ago. We believe that we have addressed all your concerns. In particular, we stated why it seems necessary to update $z$ in a coordinate manner, and we also added experimental comparisons to the existing state-of-the-art. Thus, we hope that our response will be helpful to assist you in re-evaluating our manuscript.
> Please note that the deadline of the author-reviewer discussion stage is approaching. Therefore, please let us know your further views.  If you believe we have not addressed some of your concerns, please let us know why, so that we can either realize there does exist an issue somewhere or we have a chance to clarify further.
>
>
> Best,
>
> Authors.

---

### Official Review · Reviewer_Adyx · 2021-11-02

**Correctness:** 3
**Technical Novelty And Significance:** 2
**Empirical Novelty And Significance:** 1
**Recommendation:** 3
**Confidence:** 4

**Main Review:**

While this paper has some promise, it has some major issues that prevents me from being sure of the significance of the contributions.

1. The authors assume compactness of set $\mathbb{X}$ and hence the subproblem for $x^{k+1}$ is not easy to solve even in the convex case. When $g_i$ is a convex function, this problem will be related to the proximal operator of $g_i + \delta_{\mathbb{X}_i}$ where $\delta$ is an indicator function. Even though the proximal operator of individual terms in the sum can be easy, the proximal operator of the sum is not always easy to compute.

I see this as a shortcoming of the analysis which requires compact domains. We can see in the proof of Th 4.1 (b), compactness of $\mathbb{X}$ is used to show boundedness of the sequence, from which existence of a subsequence immediately follows. Indeed, in standard analysis, we derive boundedness of the iterates, rather than assuming it.

2. The authors assume that there exists $L_g, G_g$ such that for all $s\in\partial g(x)$, one has $\|s\| \leq L_g + G_g \|x\|$. But we also have that $\| x\|$ is bounded due to compactness of $\mathbb{X}$. What is then the difference of the assumption compared to bounded subgradient assumption? The authors also mention this at the end of page 3, where they talk about almost sure boundedness, why is it not immediate from compactness of $\mathbb{X}$?

3. Can you please give more examples and also clarify more Remark 2.1. Remark 2.1 says that update of $x^k$ only requires the prox of $\phi$, why is the $\mathbb{X}$ constraint is not enforced?

4. I think the authors using their own result for the rate of deterministic method is not very fair here. Can the authors use an existing analysis for a deterministic method to compare, such as [1] or other references therein. Of course, [1] considers a problem without the weakly convex term. The authors can compare their results with $g=0$ with [1]. Of course, the assumption in [1] is different from the one in this paper and therefore the authors should make it clear difference between their assumption and the assumption of [1].

My main problem with the authors' current comparison in Remark 4.1 is the following. When we compare SGD and GD, we cannot say that we set the number of functions in SGD to be $1$ (which would give a rate $1/\sqrt{k}$ in the convex case) and use it as a rate for GD. Obviously, with GD, we can prove a much better rate and the comparison would not be fair, if we use the result of SGD with number of functions set to be $1$.

5. The experimental results are also far from being convincing. First, as my previous point, the authors need to use an existing deterministic algorithm as a reference, rather than their algorithm with $N=1$. Moreover, in the plots, we do not really see much of a difference between the full or stochastic methods.

6. The authors are missing a large number of references that studied linearly constrained problems or problems with metric subregularity, in the convex case. For example [2, 3, 4]. At the beginning of page 4, the authors talk about $g$ being null in the reference they cite. In [2, 3], g is not null and hence the authors should provide a better argument. For example, the reference that the authors cite (Zhu, Zhao, 2020) only proves almost sure subsequential convergence even for convex case, whereas [2, 3, 4] all prove almost sure sequential convergence in this convex case, which is of course much stronger.

Minor comments: 1. It will be easier to follow the theoretical results if the authors group assumptions separately. Right now, the authors say "under the assumptions in Lemma 4.1" where Lemme 4.1 says "under the assumptions of Lemma 3.1", etc., which makes it harder for the reader to follow.

[1] https://arxiv.org/pdf/1801.01994.pdf
[2] https://arxiv.org/pdf/1706.02882.pdf
[3] https://arxiv.org/pdf/2007.06528.pdf
[4] https://arxiv.org/pdf/1508.04625.pdf

**Summary Of The Paper:**

This paper develops a randomized primal-dual algorithm for solving linearly constrained problems with a nonsmooth nonconvex objective and convex constraint set. Specifically, the objective is composed of a smooth nonconvex term and a weakly convex and hence nonsmooth nonconvex term. The authors show that the limit points of the algorithm are stationary points. The authors then prove almost sure asymptotic rate of $1O(/\sqrt{k})$ and complexity of $O(\epsilon^{-2})$ for obtaining an $\epsilon$-stationary point, in expectation. Some preliminary empirical results are included.


**Summary Of The Review:**

While this paper has some promise, it has some major issues that prevents me from being sure of the significance of the contributions, such as the inaccuracy in comparison of complexities, or per iteration costs. Moreover, the assumptions seem quite strong to guarantee some important conclusions (such as boundedness of the sequence, see the remark in Th 4.1 (b)).

---

> ### Author Response · Authors · 2021-11-13
> **Reply to Reviewer Adyx**
>
> Thanks for your detailed and constructive comments.  Let us address your main criticisms on the compactness of $\mathbb{X}$ and the comparison to the existing works.  Here, compactness  is not introduced for the purpose of showing that $\{x^k\}$ is bounded. Instead,  it, together with the local uniform metric subregularity, is used to establish the (expected) sufficient decrease property of N-RPDC; see our Lemma 3.1, its following paragraph, and Section 3.3. The difficulty for showing descent is due to the $general$ linear constraint $Ax = b$ (without any assumption on $A$ and $b$) and the term $g$. The work [1] gives the first provable results of single-loop primal-dual algorithm for nonconvex optimization with general linear constraint, in which they consider problem (P) with $g\equiv 0$ and $\mathbb{X}$ being a ($compact$) box constraint.
>
> It is possible to remove the compactness assumption if $g \equiv 0$. In [2], the same authors generalized their results in [1] by showing a global dual error bound, which allows to avoid compactness of $\mathbb{X}$. However, their result requires that $g$ is null.
>
> Compactness can also be removed by considering certain ${special}$ linear constraint.  For example, suppose the variable $x$ is decomposed into $N$ blocks and consider the special conditions on the linear constraint: $\mathbb{X}_N=\mathbb{R}^{d_N}$, $g_N (x_N)\equiv0$,  $A_N$ is surjective  and
>
> $ Im(A_N)\subseteq Im((A_{1},A_{2},\ldots,A_{N-1},-b))$,  then the works [3,4] establishes convergence results for their proposed primal-dual algorithms. The work [5] you mentioned belongs to this case with $N= 2$, where their linear constraint is $Ax -z =0$ with both $z$ (treated as block $x_1$) and $x$ (treated as block $x_2$) being variables and $A$ being surjective.   However, we have to emphasize that the problem setting and proof techniques  in this line of works are very different from the general linear constraint case; see [1] for details.  We  added a comparison to the work [5] in `Related works' in the revised version.
>
>
> To our knowledge, there is not an existing way to avoid the compactness of $\mathbb{X}$ when problem (P) simultaneously has the term $g$ and the general linear constraint $Ax = b$. One possible direction is to establish a global version of the uniform metric subregularity we used.  This is left as a future direction and out of scope of the current manuscript.
>
>
> We now address your comments point-by-point.
>
> (1) When $X_i$ is a compact constraint on each coordinate $x_i\in \mathbb{R}$ (i.e., $N=d$), it must be a box constraint, which is the same as the box constraint considered in [1].  In this case, prox of $g_i + \delta_{\mathbb{X}_i}$ admits a closed-form solution when $g_i$ is an absolute value, SCAD, or MCP function.  In general, it is true that this prox is hard to compute, but the subproblem of coordinate method may still be simpler than that of the  full method, which is one of the main spirit of coordinate method.
>
> (2) We apologize for the redundant assumption. We changed it to the bounded subgradients assumption over $\mathbb{X}$; see the first paragraph of Section 1.
>
> Indeed, establishing the boundedness of the dual variable sequence $\{p^k\}$ and the auxiliary variable sequence $\{z^k\}$ is not trivial and is the main part of the proof of our Theorem 4.1.  To avoid confusion, we deleted the relevant statements about showing boundedness of the  iterates in the 3rd paragraph of `Main contribution' in the revised version.
>
> (3) Thanks for pointing this out.  We have corrected Remark 2.1 in the revision.
>
> (4) Per our above analysis, the problem setting, algorithm, and proof techniques considered in [5] are quite different from ours. Therefore, it is not very reasonable to compare our results to those of [5].
>
> We agree that we should not treat our algorithm with $N=1$ as the full method. We removed the original Remark 4.1 and added comparison to the most related work [1], even though they do not deal with the term $g$; see Remark 4.1 and 4.2 in the revised version.
>
> (5) We have changed the statements in Section 5 to make it more precise in the  revision. We have also added the experimental comparison to the algorithm proposed in [1] when $g$ is null.
>
> (6) Thanks so much for providing us these useful references. We have mentioned these works properly in `Related works' in the revised version.  BTW, what we used is the $\mbox{\it uniform}$ metric subregularity, which is essentially different  from the notion of metric subregularity.
>
> Thanks again for providing the constructive comments, which helped  us a lot to improve the quality of our work.
>
> [1] Zhang \& Luo (2020) SIOPT,30(3), 2272-2302.
>
> [2] Zhang \& Luo (2020), arXiv preprint arXiv:2006.16440.
>
> [3] Bolte et. al. (2018) MOR, 43(4), 1210-1232.
>
> [4] Hong et. al. (2016) SIOPT, 26(1), 337-364.
>
> [5] Bot \& Nguyen (2020) MOR, 45(2), 682-712.

---

> > ### Comment · Reviewer_Adyx · 2021-11-26
> > **Discussion**
> >
> > I thank the authors for their responses.
> >
> > Unfortunately, my biggest issue still remains on the comparison with existing works/utility of the results: It is not clearly justified if randomization is helpful in this setting.
> >
> > To be specific, Remark 4.1 is now changed by authors to not argue a complexity comparison but to say that "the rate is the same as deterministic results, but randomized methods are more suitable for large scale problems". Unfortunately, this generic claim is not always true and this is why it is important for the authors to identify problems where the randomization is indeed useful. For example, even in the simplest case of random coordinate descent (RCD) (Nesterov, 2012), it depends on the comparison of the one iteration cost of RCD vs one iteration cost of GD on whether RCD improves the overall complexity compared to GD, or not. The reason is that even though per iteration cost can be cheaper for RCD, we need to make more iterations than GD in most cases, then it depends on the "overall cost to get $\epsilon$ accuracy" to see which one is preferable. In convex case, we know many interesting cases when this is the case, but this is not a given and for the problems where the cost of RCD is not cheap enough, GD is preferable and hence we cannot say that "randomization is better for large scale problems, in general".
> >
> > In view of this, right now I am not convinced that this paper shows:
> >
> > - the randomized algorithm is better in theory, or
> > - the randomized algorithm is better in practice.
> >
> > For the latter, the experimental results are too simplistic and we do not really see any significant difference between the methods. For the former, as I wrote above, only generic statements are given such as "randomization is useful in large scale problems", which is not sufficient.
> >
> > Moreover, I see that the authors have difficulty comparing their assumption (depending on metric subregularity), and the assumptions for existing results depending on KL-inequality (or other assumptions, looking into the comments of 6TRm). I understand that it is hard to compare different assumptions, but it is necessary, especially for such technical works. **It is the responsibility of the authors to position their work/results within the already existing results**. Otherwise, having different works, claiming similar results, without comparing to each other makes it almost impossible for the readers to follow the literature. The current discussion the authors added comparing to the work of Bot and Nguyen just restates their results and assumptions, rather than explaining how their assumption differs from the previous work and which important applications each assumption covers.
> >
> > Overall, I cannot recommend acceptance due to these major remaining concerns, and I recommend the authors to take the time to clarify these points for revising their paper:
> >
> > - clarify clearly when the new method with randomization improves over the existing methods without randomization, and/or illustrate this point in practice
> > - understand and clearly compare the different assumptions made in different works and how their assumption is stronger/weaker and what the implications are, in terms of applications.

---

> > > ### Author Response · Authors · 2021-11-27
> > > **Reply on "comparison of assumptions/error bound conditions"**
> > >
> > >
> > > Let us now address your another comment on "comparison of assumptions/error bound conditions".
> > >
> > > We are afraid that you misunderstood why and how we use the $\mbox{\textbf{uniform metric subregularity}}$---please do not miss the word $\mbox{\textbf{uniform}}$ since this error bound condition is $\mbox{\textbf{fundamentally}}$ different from the (classical) metric subregularity condition.
> > > The uniform metric subregularity is used for establishing the sufficient decrease property of our N-RPDC.  The reason that we need an error bound condition to establish a descent property is due to our relatively hard problem setting, i.e., the general linear constraint $Ax = b$ and the nonsmooth nonconvex term $g$. Thanks to this error bound condition, we can derive a useful upper bound for $||x(z,p)-x(z)||$ $\mbox{\textbf{uniformly over $z$}}$, which allows us to show the descent property of N-RPDC in expectation in terms of the Lyapunov function $\Lambda$; see the paragraph below Lemma 3.1 for explanation. Then, the convergence of N-RPDC  can be established. By contrast, if one considers certain special linear constraint, e.g., the setting of (Bot and Nguyen, 2020), the sufficient decrease property can be obtained as a natural property of the algorithm without using any error bound condition.
> > >
> > > In addition, we have the following to clarify.
> > >
> > > $\bullet$ There is not existing convergence result for solving our problem setting by utilizing the KL inequality.
> > >
> > > $\bullet$ The uniform metric subregularity was newly introduced very recently  in (Kruger and Cuong, 2020), which is a $\mbox{\textbf{parameter error bound}}$ defined on some parameter set-valued mapping, e.g., defined on the mapping $\partial\mathcal{L}_{\gamma}^+$ uniformly over the parameter $z$ in our manuscript. It is used here for establishing descent property of N-RPDC as explained above. By contrast, the KL inequality is an error bound condition used for establishing strong sequential convergence (and rate) for algorithms that already possess the descent property. Thus, the usage of these two conditions are very different, and (after replying to Reviewer 6TRm) we realized it is very unlikely that one can replace the uniform metric subregularity here with the KL inequality.
> > >
> > > $\bullet$ The pioneering work (Zhang and Luo, 2020) uses the  dual error bound condition for establishing the descent property of their algorithm. Again, let us emphasize that the use of an error bound condition to establishing a sufficient decrease property in (Zhang and Luo, 2020) is due to their hard problem setting (i.e., the general linear constraint).  However, such an error bound is no longer applicable for our problem (P) due to the term $g$.  The deep reason is that the dual error bound is established by using the well-known Hoffman's error bound, which is not shown to be true for nonsmooth nonconvex functions yet.   Instead, we identified the uniform metric subregularity for our problem (P) with term $g$, which is crucial for analyzing our N-RPDC.
> > >
> > > We hope that our response is satisfactory to you. We definitely welcome additional insightful comments from you (if any) so that we can try to clarify further.
> > >
> > > $\mbox{\bf Reference}$
> > >
> > > Bot, R. I., \& Nguyen, D. K. (2020). The proximal alternating direction method of multipliers in the nonconvex setting: convergence analysis and rates. Mathematics of Operations Research, 45(2), 682-712.
> > >
> > > Kruger, A. Y., \& Cuong, N. D. (2021). Uniform Regularity of Set-Valued Mappings and Stability of Implicit Multifunctions. Journal of Nonsmooth Analysis and Optimization, 2.
> > >
> > > Zhang, J., \& Luo, Z. Q. (2020). A proximal alternating direction method of multiplier for linearly constrained nonconvex minimization. SIAM Journal on Optimization, 30(3), 2272-2302.

---

> > > ### Author Response · Authors · 2021-11-27
> > > **Reply on "whether randomized coordinate update help"**
> > >
> > > Thanks for your additional comments. Let us clarify further on "whether randomized coordinate update help".
> > >
> > >
> > > $\bullet$ $\mbox{\emph{Complexity comparison.}}$ Let us first clarify that there is no existing coordinate-type algorithm for our problem setting, neither cyclic coordinate nor randomized coordinate method. Thus, let us understand your concern "It is not clearly justified if randomization is helpful in this setting" as "It is not clearly justified if randomized coordinate update is helpful in this setting".
> > >
> > > In our revised Remark 4.1, we roughly and quickly stated that our complexity result has the same order as that of (Zhang and Luo, 2020) since we  want to avoid such an 'unreasonable' comparison.  First, our complexity result is in expectation, while the complexity result in (Zhang and Luo, 2020) is deterministic.  It is not that reasonable to compare these two types of complexity results since they are different in nature.  Secondly, our result can deal with problem (P) with the nonsmooth nonconvex term $g$, while the work (Zhang and Luo, 2020) cannot.  Thirdly, both are worst-care complexity results. The difference may come from, e.g.,  the different proof techniques. A better worst-case complexity result is not sufficient to conclude an algorithm has better property.
> > >
> > >
> > > $\bullet$ $\mbox{\emph{Suitability for large-scale problems.}}$ We totally agree with you that randomized coordinate method is not always faster than the full method. However, randomized coordinate algorithm often has its own merit, as clarified in (Nesterov, 2012; Richtarik and Takac, 2014). When the optimization problem is large-scale (measured as the dimension of the variable), the computation of a single function value or gradient can be prohibitive. For example, suppose the dimension of the space of variables is larger than the available memory. In that case, forming a gradient or even evaluating the function value may be impossible, and hence the full methods will not work.  Moreover, the nature and structure of data describing the problem may be an obstacle in using full methods. For example, if the problem data is distributed in space and in time, then it may be necessary to work with whatever data is available.  In all the above situations, It appears that a very reasonable approach to solving such problems is to use randomized coordinate-type methods.
> > >
> > > To be more accurate, we will change the last sentence in Remark 4.1 to "which is more suitable for $\mbox{\textbf{certain}}$ modern large-scale problems" in the next version.
> > >
> > >
> > > Let us answer your another comment on "comparison of assumptions" in the the following reply.
> > >
> > > $\mbox{{\bf Reference}}$
> > >
> > > Zhang, J., \& Luo, Z. Q. (2020). A proximal alternating direction method of multiplier for linearly constrained nonconvex minimization. SIAM Journal on Optimization, 30(3), 2272-2302.
> > >
> > > Nesterov, Y. (2012). Efficiency of coordinate descent methods on huge-scale optimization problems. SIAM Journal on Optimization, 22(2), 341-362.
> > >
> > > Richtarik, P., \& Takac, M. (2014). Iteration complexity of randomized block-coordinate descent methods for minimizing a composite function. Mathematical Programming, 144(1), 1-38.

---

> > > > ### Comment · Reviewer_Adyx · 2021-11-28
> > > > **Replies for discussion**
> > > >
> > > > > *" First, our complexity result is in expectation, while the complexity result in (Zhang and Luo, 2020) is deterministic. It is not that reasonable to compare these two types of complexity results since they are different in nature."*
> > > >
> > > > This is not an unreasonable comparison and, to my knowledge, is always done for random coordinate descent or stochastic gradient (or variance reduced) methods. The authors can see Nesterov, 2012 for RCD and Le Roux, Schmidt, Bach 2013 for SAG and varience reduction and virtually all the followup papers on these two lines. Even if the comparison is unfair, it favors random methods since the result in expectation is weaker than deterministic. If the random method does not have better complexity even in this case, then it is not reasonable to expect it can be better in any other case.
> > > >
> > > >  > *"A better worst-case complexity result is not sufficient to conclude an algorithm has better property."*
> > > >
> > > > This is a correct and philosophical remark. On the other hand, most of the optimization literature compare the algorithms w.r.t. worst-case bounds. Moreover, the authors cannot show that their method is better in "average case" or "best case", so I'm not sure how this is relevant.
> > > >
> > > > > *"For example, suppose the dimension of the space of variables is larger than the available memory"*
> > > >
> > > > In this setting, the methods in this paper also does not apply. Note that RCD methods including this paper has to store the full primal variable ($x, z$ in this paper). If the authors believe they can run their algorithm without storing the these variables, they have to show how because it is not clear in general for RCD methods.
> > > >
> > > > > *"For example, if the problem data is distributed in space and in time, then it may be necessary to work with whatever data is available."*
> > > >
> > > > I cannot see any results in this paper, specified to the distributed setting. Saying that a method and results are applicable to distributed setting is one thing, proving it rigorously is another. If the authors want to stress this setting, I recommend them to prove it clearly in their revisions.
> > > >
> > > > > *"Let us first clarify that there is no existing coordinate-type algorithm for our problem setting, neither cyclic coordinate nor randomized coordinate method"*
> > > >
> > > > This,  by itself is not a justification for the significance of work, unless the authors can prove coordinate-type algorithm helps in theory and or practice. However, both are not clear from the results of this work, at this point.

---

> > > > > ### Author Response · Authors · 2021-11-29
> > > > > **Replies for reviewer Adyx**
> > > > >
> > > > > Thanks for your time and insightful comments.
> > > > >
> > > > > $\mbox{\textbf{1. Complexity comparison}}$
> > > > >
> > > > > In order to compare the complexity results between N-RPDC and that of [1], let us assume $g \equiv 0$,  otherwise  the result in [1] does not apply.  In this case,  on the one hand, the work [1] can choose the dual step size $\eta = \eta_0$ to be some constant, while we have to choose $\eta \approx \frac{\eta_0}{N}$. We note that the dual step size of the coordinate-type primal-dual methods are usually chosen as $\frac{1}{N}$ times smaller than the full primal-dual method's dual step size; see, e.g., [2,3,4,5]. On the other hand, our step sizes $\alpha_x$ and $\alpha_z$ for primal variables $x$ and $z$ can be selected the same order as that of [1].  As a consequence, our complexity result will be $N$ times worse that of [1]. However, as we explained in Remark 2.1, the computational complexity of N-RPDC can be $\frac{1}{N}$ times cheaper than the full primal-dual method [1].  Overall, if we consider $N$ consecutive iterations of N-RPDC and one iteration of the full primal-dual method in [1] as one epoch, the complexity bounds of these works are $O(\sqrt{{1}/{t}})$, where $t$ counts the number of epochs and the big-O is independent of $N$. Therefore, the two complexity bounds are comparable. We will add the above comparison in our next revision.
> > > > >
> > > > > This may raise another question.  Why we need to concern coordinate-type primal-dual method, provided that we cannot improve the theoretical complexity result of the full primal-dual method. We will try to address this concern in the response below.
> > > > >
> > > > > $\mbox{\textbf{2. Practical motivations for N-RPDC}}$
> > > > >
> > > > > $\bullet$ $\mbox{\emph{Computational memory for large-scale problems}}$. We apologize for the confusing explanation. Yes.  N-RPDC also has to store the full primal and dual iterates.  However, what we meant to say is that each iteration of N-RPDC is much cheaper and usually needs much less memory for implementing updates.  Note that memory can be classified into data storage (ROM-like) and computational memory (RAM-like).  The advantage of N-RPDC is that each subproblem is a lower-dimensional (even scalar) minimization problem, and it can typically be solved more easily than the full primal-dual method, i.e., it requires less computational memory. Thus, in the case where the computational power is limited, it can be preferable to utilize N-RPDC rather than the full primal-dual method. We will add this motivational statement for N-RPDC accurately in the next revision.
> > > > >
> > > > > $\bullet$ $\mbox{\emph{Data distribution in time}}$.  Let us again apologize for the confusing explanation.  What meant to say is that if the data is coming in time (we also need to assume it comes in an i.i.d. uniform manner due to the sampling scheme of our algorithm), N-RPDC may  be preferable. If we want to apply the full primal-dual method in such a situation, we  have to  wait for the entire data set to arrive before the optimization process is started.  By contrast, N-RPDC can implement updates in this case whenever data is available.  We will also add this motivational statement for N-RPDC accurately in the next revision.
> > > > >
> > > > > We guess the confusion caused  is due to the sentence that  "when the data is distributed in space", which relates to distributed coordinate-type methods. We agree with you that we need new analysis for this type of distributed coordinate methods. We will point out this direction in our next revision.
> > > > >
> > > > > $\mbox{\textbf{3. Issues of claiming 'the first work'}}$
> > > > >
> > > > > We agree with you that claiming our work is the first one in this setting is not a justification for significance.  Though we clarified that the theoretical complexity of N-RPDC is almost the same as that of the full primal-dual method  [1] if $g\equiv 0$ (see above), we tried to motivate our study from practical points for view (say, computational memory and data distribution in time). We hope that these points can at least serve as the motivations of our coordinate method, N-RPDC.
> > > > >
> > > > > $\mbox{\textbf{4. Deeply thanks}}$
> > > > >
> > > > > Finally, we deeply thank the reviewer for your time and very insightful comments, which helped us understand some fundamental and motivational concepts of the coordinate method better.
> > > > >
> > > > > References
> > > > >
> > > > > [1] Zhang, J., & Luo, Z. Q. (2020). SIAM Journal on Optimization, 30(3), 2272-2302.
> > > > >
> > > > > [2] Gao, X., Xu, Y. Y., & Zhang, S. Z. (2019). Journal of the Operations Research Society of China, 7(2), 205-250.
> > > > >
> > > > > [3] Xu, Y., & Zhang, S. (2018). Computational Optimization and Applications, 70(1), 91-128.
> > > > >
> > > > > [4] Xu, Y. (2021). Informs Journal on Optimization, 3(1), 89-117.
> > > > >
> > > > > [5] Zhu, D., & Zhao, L. (2020, November). International Conference on Machine Learning (pp. 11619-11628). PMLR.

---

> ### Author Response · Authors · 2021-11-26
> **To Reviewer Adyx**
>
> Dear Reviewer,
>
> Thanks again for reviewing our manuscript and for your constructive comments. It has been 13 days since we submitted our response.   Please note that the deadline of the author-reviewer discussion stage is approaching. We are writing to kindly request your feedback on our rebuttals.  Since you had  questions on our compactness assumption and on our comparison to the existing works in your review,  we stated carefully the reason why we impose such a compactness assumption (due to the relatively hard problem setting of our manuscript), and we also revised our comparison to the existing literature according to your comments.  We believe that our response addressed your concerns and will be helpful to assist you in re-evaluating our manuscript.  We hope that you would let us know if our response was satisfactory. If you believe we have not addressed some of your concerns, please let us know why, so that we can either realize there does exist an issue somewhere or we have a chance to clarify further.
>
>
>  Best,
>
>  Authors.

---

### Official Review · Reviewer_6TRm · 2021-11-03

**Correctness:** 3
**Technical Novelty And Significance:** 2
**Empirical Novelty And Significance:** 3
**Recommendation:** 6
**Confidence:** 2

**Main Review:**

strengths:
S1. This paper is well-written. It illustrates the basic idea of the coordinate descent method for solving linearly constrained nonconvex nonsmooth problems in a clear way.
S2. The proposed N-RPDC algorithm is novel. Using the local uniform metric subregularity property, the authors establish the global convergence of the proposed algorithm and prove that any limit point of the algorithm is a clustering point. I find the proofs of this paper are rather technical and nontrivial.
S3. The authors provide nice explanations and discussions for their algorithm design and theoretical analysis.


weaknesses
W1. The proposed algorithm involves several unknown parameters (e.g., rho_g, sigma), these parameters could greatly affect the practical performance of the coordinate descent methods. The authors do not discuss the practical choice of these parameters in the experiment section.

W2. In the sufficient decrease properties in lemma 4, the parameters kappa and delta are unknown, it is not known how to choose these parameters to guarantee the global convergence of the algorithm.

W3. The convergence analysis of ALM for solving nonconvex problems has been well-studied using the Kurdyka-Lojasiewicz inequality assumption. It is not clear whether the new local uniform metric subregularity assumption is weaker than the KL inequality assumption.


minor issues:
M1. The parameter beta_K is not defined in Lemma 3.1. I think it is the strong convexity parameter of the Bregman distance function.
M2. It seems to me that solving the auxiliary problem reduces to the inertial proximal or Nesterov's momentum strategy, which has been wildly used in solving the minimax optimization problems. It will be helpful to make discussions on this point.



**Summary Of The Paper:**

This paper proposes a randomized primal-dual coordinate method for solving linearly constrained nonsmooth nonconvex optimization problems. At each iteration, this method only selects a primal variable to update randomly. The proposed algorithm can solve large-scale problems since the computational complexity in each iteration is very cheap compared with the full gradient methods. The authors prove that any cluster point of the sequence of iterates is almost surely a stationary point. Some experiments on the non-PSD kernel SVM problem and the linearly constrained Lasso problem show that it is practical.



**Summary Of The Review:**

See above.

---

> ### Author Response · Authors · 2021-11-13
> **Reply to Reviewer 6TRm**
>
> We appreciate your positive evaluation on our paper.
>
> W1-2: We argue that  the issue of unknown parameters exists widely in the theoretical analysis for algorithms, e.g.,  the gradient Lipschitz parameter, strongly convex parameter, bounded variance parameter, etc,  in the convergence theory of GD and SGD. Though our theory involves  these unknown parameters, N-RPDC works reasonably well and exhibits nice convergence properties in practice with less carefully hand-tuned step sizes.
>
> W3: KL framework is often used for establishing strong convergence results for algorithms that possess a certain sufficient decrease property. Here, the local uniform metric subregularity is used to establish the (expected) sufficient decrease property of N-RPDC. Thus, the usage of these two tools are quite different. The reason that we need an error bound condition for establishing descent property of N-RPDC is due to the relatively hard problem setting considered in this paper; see also our reply to Reviewer Adyx.  It is not clear what is the relationship between KL property and the local uniform metric subregularity, which is indeed an interesting topic to discuss in the future.
>
> M1: Yes, it is the strong convexity parameter of the Bregman function.  We have added the definition of $\beta_K$ in the second-to-last paragraph above Remark 2.1 in the revision.
>
> M2: Thanks for pointing the possible relations between the auxiliary problem and the inertial- or the momentum-type algorithms. In the future, it is indeed quite interesting to see if it is possible to keep all the convergence theory even without introducing the auxiliary variable $z$.
>
> We hope that these revisions and responses are satisfactory to you.

---

> ### Author Response · Authors · 2021-11-26
> **To Reviewer 6TRm**
>
> Dear Reviewer,
>
> Thanks again for reviewing our manuscript and for your positive evaluation. We are writing to kindly request your feedback on our rebuttals, which we submitted 13 days ago. We believe that we have addressed all your concerns. Specifically, we restate the hard problem setting of our manuscript, which we hope will be helpful to assist you in re-evaluating our manuscript.    Please note that the deadline of the author-reviewer discussion stage is approaching. Therefore, please let us know your further views.  If you believe we have not addressed some of your concerns, please let us know why, so that we can either realize there does exist an issue somewhere or we have a chance to clarify further.
>
>
> Best,
>
> Authors.

---

### Decision · Program_Chairs · 2022-01-20

**Decision:**

Reject

**Comment:**

Dear Authors,

The paper was received nicely and discussed during the rebuttal period. However, the current consensus suggests the paper requires another round of revisions before it gets accepted.

In particular:

- it is not clear if the new method with randomization improves over the deterministic methods, either in theory and practice.
- it is not clear how the assumptions made in this work compare to the existing ones and what the implications are, in terms of applications.

Reviewers were not satisfied by the replies received during the rebuttal period.
One reviewer stated that the argument "first coordinate method for this setting" is valid, but not sufficient to justify publication at this stage.

Best
AC